# MYPT1-PP1β phosphatase negatively regulates both chromatin landscape and co-activator recruitment for beige adipogenesis

Protein kinase A promotes beige adipogenesis downstream from β-adrenergic receptor signaling by phosphorylating proteins, including histone H3 lysine 9 (H3K9) demethylase JMJD1A. To ensure homeostasis, this process needs to be reversible however, this step is not well understood. We show that myosin phosphatase target subunit 1- protein phosphatase 1β (MYPT1-PP1β) phosphatase activity is inhibited via PKA-dependent phosphorylation, which increases phosphorylated JMJD1A and beige adipogenesis. Mechanistically, MYPT1-PP1β depletion results in JMJD1A-mediated H3K9 demethylation and activation of the *Ucp1* enhancer/promoter regions. Interestingly, MYPT1-PP1β also dephosphorylates myosin light chain which regulates actomyosin tension-mediated activation of YAP/TAZ which directly stimulates *Ucp1* gene expression. Pre-adipocyte specific *Mypt1* deficiency increases cold tolerance with higher *Ucp1* levels in subcutaneous white adipose tissues compared to control mice, confirming this regulatory mechanism in vivo. Thus, we have uncovered regulatory cross-talk involved in beige adipogenesis that coordinates epigenetic regulation with direct activation of the mechano-sensitive YAP/TAZ transcriptional co-activators.

Cold stress is a major threat to mammals; thus, thermoregulation is essential for survival. Uncoupling protein 1 (UCP1) is crucial for thermoregulation in brown adipose tissue (BAT) where its expression in the inner mitochondrial membrane drives the uncoupling of cellular respiration from ATP synthesis to produce heat. In addition to BAT, mammals also induce the appearance of brown-like thermogenic adipocytes in subcutaneous white adipose tissue (scWAT) and this also through the uncoupling activity of UCP1. This metabolic adaptation is referred to as beiging of scWAT and it is induced by selective environmental stimuli, including chronic cold stress, hormonal activation, and pharmacological treatments[1–8].

Beta adrenergic receptor (β-AR) signaling in scWAT plays a central role in beige adipogenesis in response to norepinephrine (NE) secreted from sympathetic nerves. The resulting increase in intracellular cyclic

AMP (cAMP) leads to elevated protein kinase A (PKA) activity that drives coordinated changes in gene expression through transcriptional and epigenetic programs. Despite a fundamental understanding of the mechanism for how PKA signaling drives thermogenesis, there are many gaps that remain elusive. One of the PKA substrates involved in thermogenesis in BAT is Jumonji-C domain-containing 1a (JMJD1A, also known as JHDM2A or KDM3A)[9–11], which is a demethylase specific for mono- and di-methylated versions of H3K9 (H3K9me1/me2)[1,9]. JMJD1A is also important for beige adipogenesis through a concerted multistep process[1]: (i) β-AR activation results in PKA-dependent phosphorylation of JMJD1A at S265. This phosphorylation nucleates the formation of a complex between JMJD1A with PRDM16, PGC1α, and PPARγ. (ii) This complex is then recruited to enhancer/promoter regions of thermogenic genes (1st step, cold sensing). (iii) pS265-JMJD1A removes

✉e-mail: matsumura-y@lsbm.org; jmsakai@med.tohoku.ac.jp

repressive H3K9me2 to induce thermogenic gene expression (2nd step, epigenetic re-writing). The beiging process is rapidly reversible[12], suggesting that dephosphorylation of JMJD1A might be a key regulated step as well. Therefore, an unidentified phosphatase(s) that antagonizes PKA phosphorylation of JMJD1A could be a molecular target for the transient nature of the beiging process in scWAT.

Yes-associated protein 1 (YAP1) and its paralog transcriptional co-activator with PDZ-binding motif (TAZ) function as mechanotransducers of mechanical signals into the nucleus[13]. On activation, cytoplasmic YAP/TAZ translocate into the nucleus and interact with TEA domain (TEAD) transcription factors to activate gene expression[14]. YAP/TAZ activity is regulated by actomyosin tension, which is induced by phosphorylation of myosin regulatory light chain (RLC), thereby mediating the nuclear transduction of mechanical and cytoskeletal signaling pathways[13,15]. Of note, recent studies have shown that actomyosin tension regulates *Ucp1* expression through YAP/TAZ recruitment to the *Ucp1* enhancer in brown adipocytes[16].

Here, we identify the regulatory and catalytic subunits of myosin phosphatase (MP): myosin phosphatase target subunit 1 (MYPT1) and protein phosphatase 1 catalytic subunit beta (PP1β), respectively, as a phosphatase that dephosphorylates pS265 of JMJD1A. Consistent with this process being a key regulatory step, we demonstrate that MYPT1-PP1β depletion results in an increased and sustained thermogenic capacity of beige adipocytes. We also show that MYPT1-PP1β dephosphorylates RLC which is key to regulating the actomyosin-sensitive YAP/TAZ pathway that mediates transcriptional activation of *Ucp1*. Importantly, upon β-AR activation, MYPT1-PP1β activity is inhibited through T694 phosphorylation of MYPT1 by PKA, enabling efficient 'on-off' switch of cold-induced phosphorylation. Therefore, our studies reveal that MYPT1-PP1β is a crucial regulator of beige adipogenesis, and the underlying mechanism involves the coordinate regulation of both epigenetic and direct transcriptional co-factor pathways to amplify the response to by β-AR signaling.

## Results
### Identification of MYPT1-PP1β as a pS265-JMJD1A phosphatase
To search for phosphatase(s) of pS265-JMJD1A (Fig. 1a), JMJD1A and its associated proteins were immunoprecipitated through tandem affinity purification. 3T3-L1 pre-adipocytes were retrovirally transduced with an empty vector or with a vector expressing FLAG-tagged human JMJD1A, and JMJD1A was then subjected to two rounds of immunoprecipitation, first with anti-FLAG and then with anti-human JMJD1A antibodies. Proteins eluted from the second affinity step were subjected to liquid chromatography tandem mass spectrometry (LC-MS/MS) (Fig. 1b). Two independent proteomic analyses (experiments 1 and 2) using JMJD1A co-immunoprecipitates identified components of the myosin phosphatase: MYPT1 and PP1β as potential interacting phosphatase proteins of JMJD1A (Fig. 1c and Supplementary Data 1). The interaction of MYPT1 and JMJD1A was confirmed by direct co-immunoprecipitation analysis in NIH-3T3 cells pre-treated with phosphodiesterase inhibitor 3-isobutyl-1-methylxanthine (IBMX) which raises intracellular cAMP (Fig. 1d). To determine whether MYPT1-PP1β might be responsible for the dephosphorylation of pS265-JMJD1A, immortalized pre-adipocytes from scWAT (im-scWAT) were analyzed by immunoblotting using the pS265 specific antibody. Pretreatment of the cells with calyculin A, a PP1 and PP2A inhibitor, markedly increased the phosphorylation level of JMJD1A (Fig. 1e). We next knocked down MYPT1 using small interfering RNA (siRNA) (Fig. 1f and shown later in Fig. 1i) which also significantly increased pS265-JMJD1A (Fig. 1f). These results suggest that MYPT1-PP1β is a candidate phosphatase for pS265-JMJD1A.

### MYPT1-PP1β negatively regulates beige adipogenesis
To investigate the role of MYPT1-PP1β in beige adipogenesis, we performed knockdown studies in primary pre-adipocytes isolated from the stromal vascular fraction (SVF) of scWAT of wild type (*WT*) mice. A control siRNA (si-Ctrl) or an siRNA targeting *Mypt1* (si-*Mypt1*) was transfected into the pre-adipocytes (Supplementary Fig. 1a, b) and we performed RNA-sequencing (RNA-seq) on total RNA prepared at (day 0), or following beige adipocyte differentiation (day 8) (Fig. 1g). Among 3,356 beige-selective genes (fold change (FC) of reads per kilobase million (RPKM) (si-Ctrl day 8/si-Ctrl day 0 > 1.5)), 115 genes (Supplementary Table 1) were increased by MYPT1 depletion (FC of RPKM (si-*Mypt1* day 8/si-Ctrl day 8 > 1.5)). These include thermogenic genes such as *Ucp1*, *Fabp3*, *Cidea*, *Cox8b*, *Otop1*, *Elovl3*, and *Cpt1b*. Kyoto Encyclopedia of Genes and Genomes (KEGG) pathway analysis revealed that these 115 genes were associated with pathways such as PPAR signaling (Fig. 1h, top). In addition, Gene Ontology (GO) enrichment analysis showed that these genes were associated with GO terms such as regulation of lipid catabolism (Fig. 1h, bottom). We confirmed several of the affected genes from the RNA-seq results by quantitative PCR (qPCR) (Fig. 1i, Supplementary Fig. 1b, c). Oil red O staining showed that lipid accumulation (i.e., general adipogenesis) is not affected by MYPT1 depletion (Supplementary Fig. 1b, c, inset), indicating MYTP1 regulates specific thermogenic genes during adipogenesis. *Ucp1* expression was also significantly increased to a far higher level in beige adipocytes from MYPT1-depleted im-scWAT cells compared to si-Ctrl derived cells in a time- and dose-dependent manner by isoproterenol (ISO), a non-selective β-AR agonist (Supplementary Fig. 1d). The effect was selective as knockdown of PP1β by siRNA targeting *Ppp1cb* (si-*Ppp1cb*) resulted in altered thermogenic gene activation but there was no effect following knockdown of either PP1α (si-*Ppp1ca*) or PP1γ (si-*Ppp1cc*) (Fig. 1j, Supplementary Fig. 1e–g). Moreover, double knockdown of MYPT1 and PP1β further induced *Ucp1* expression compared to single MYPT1 knockdown (Fig. 1k). As the specificity of myosin phosphatase function is mediated through MYPT1[17,18], we hereafter focused on investigating MYPT1. The results in Fig. 1l show that MYPT1 depletion also increased NE-stimulated oxygen consumption rate (OCR). To examine a connection with JMJD1A and MYPT1, we combined the RNA-seq data in Fig. 1g with JMJD1A chromatin immunoprecipitation sequencing (ChIP-seq) data in im-scWAT differentiated beige adipocytes[1]. JMJD1A binding was enriched near the genes that were increased following MYPT1 depletion compared with genes that were either downregulated or unaffected (Fig. 1m). Since JMJD1A controls thermogenic program through histone demethylation, we measured H3K9me2 levels on enhancer/promoter regions of *Ucp1*, *Cidea*, and *Ppara* genes in MYPT1-depleted cells. During beige adipogenesis, H3K9me2 levels were reduced on these regions (Supplementary Fig. 1h) as we previously reported[1]. Importantly, MYPT1 and PP1β depletion resulted in further reduction of H3K9me2 levels at these same regions (Fig. 1n). Together, these data indicate that MYPT1-PP1β negatively regulates beige adipogenesis through the dephosphorylation of pS265-JMJD1A, leading to the modulation of H3K9me2 (Fig. 1o).

### MYPT1 targets RLC phosphorylation during beige adipogenesis
JMJD1A is an epigenetic modifier that changes chromatin architecture but is not a transcription factor. Inactivation of MYPT1-PP1β resulted in marked *Ucp1* induction to a level much higher than we initially expected. This promoted us to hypothesize that an additional signaling pathway, in addition to the epigenetic program we identified in our previous study[1], might be targeted by MYPT1-PP1β. Therefore, we performed two distinct phosphoproteomic analyses in im-scWAT cells using both label-free and isobaric tag labeling approaches. These two approaches revealed that phosphorylation levels of 64 and 33 phosphopeptides were markedly increased by MYPT1 depletion, respectively (Fig. 2a, b, Supplementary Fig. 2a). The myosin regulatory light chain (RLC) was identified by both approaches (Fig. 2a, b). When all the potential phospho-proteins were subjected to KEGG pathway analysis, several of the pathways contained the RLC gene (Fig. 2c, d). Interestingly, T18/S19 phosphorylation of RLC (Supplementary Fig. 2b) is

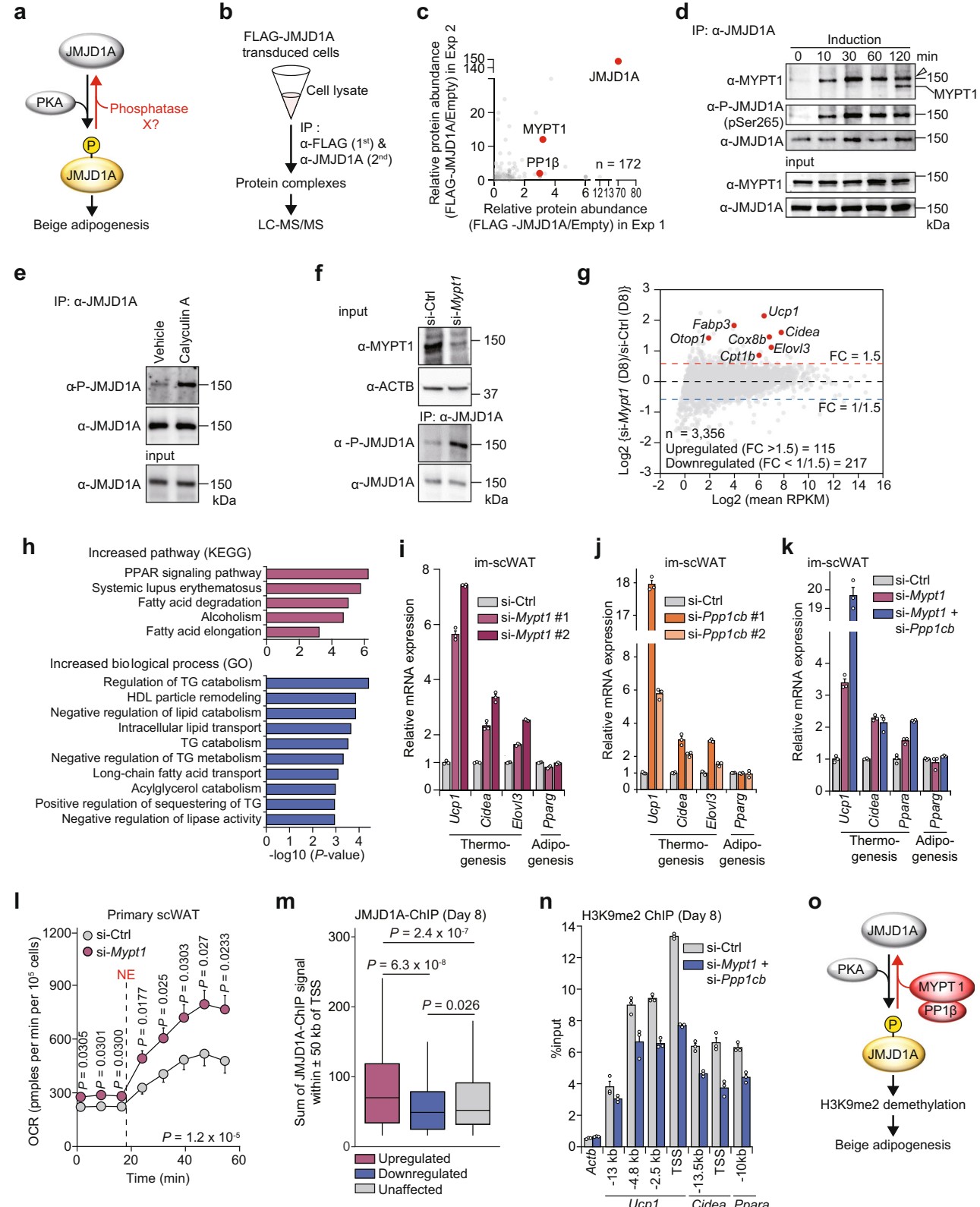

crucial for the generation of actomyosin tension, which is important for *Ucp1* regulation by the hippo pathway transcriptional co-activators YAP and TAZ in brown adipocytes[16]. Thus, we hypothesized that MYPT1-PP1β might couple the epigenetic regulation of *Ucp1* by JMJD1A with YAP/TAZ regulation by RLC induced actomyosin tension and this might explain the very high level of *Ucp1* induction by MYPT1 depletion. To test this hypothesis, we first, performed double knockdown of

MYPT1 and myosin light chain kinase (MYLK), a kinase responsible for the phosphorylation of RLC[18,19] (Supplementary Fig. 2c). MYPT1 depletion induced both thermogenic gene mRNA and RLC phosphorylation as expected, and this was reduced by additional depletion of MYLK (Fig. 2e, f, and Supplementary Fig. 2d), suggesting that MYPT1-PP1β negatively regulates *Ucp1* expression by antagonizing MYLK-mediated RLC phosphorylation. To further investigate the importance of RLC

**Fig. 1 | MYPT1-PP1β dephosphorylates pS265-JMJD1A and regulates beige adipogenesis via epigenetic rewriting. a** The phosphatase of pS265-JMJD1A is unknown. **b** Identification of JMJD1A binding proteins by LC-MS/MS. **c** Proteome scatter plot in (**b**). *X*-axis and *Y*-axis, relative protein abundance (spectrum count) in experiments 1 and 2, respectively. Identified peptide number (*n*). (**d**) MYPT1 and P-JMJD1A co-immunoprecipitation from NIH-3T3 cells after adipogenic induction with IBMX-containing differentiation cocktail. MYPT1 immunoblotting was performed without antibody stripping after P-JMJD1A immunoblotting, resulting in the detection of both MYPT1 and P-JMJD1A (arrowhead). **e, f** P-JMJD1A immunoblotting following immunoprecipitation with anti-JMJD1A antibody from day 0 im-scWAT treated with calyculin A in (**e**) or transfected with si-Ctrl or si-*Mypt1* #2 in (**f**). **g** RNA-seq MA plot of beige-selective genes in day 8 primary scWAT cultures of *WT* mice transfected with si-Ctrl or si-*Mypt1* #2. *X*-axis, log2 (mean RPKM). Y-axis, log2 (FC RPKM). The numbers of total (*n*), upregulated, and downregulated beige-selective genes following MYPT1 depletion. **h** KEGG and GO pathway analysis of upregulated genes in (**g**). **i, j** Thermogenic and general adipogenic gene mRNA levels in day 8 im-scWAT transfected with indicated siRNA. **k** Thermogenic and general adipogenic gene mRNA levels in day 8 im-scWAT transfected with si-Ctrl, si-*Mypt1* #2, or si-*Mypt1* #2 + si-*Ppp1cb* #1. **l** NE-stimulated increase in OCR of day 8 primary scWAT cultures of *WT* mice transfected with si-Ctrl or si-*Mypt1*. Data are mean ± SEM of five technical replicates. Two-way repeated measures ANOVA, followed by a post hoc Student's *t* test. **m** Box plot showing JMJD1A binding signal near unaffected, upregulated, and downregulated genes (554, 831, and 37,976 peaks, respectively) by MYPT1 depletion in (**g**). Box shows median and first and third quartiles. Genome-wide analyses were performed once based on JMJD1A ChIP-seq data set in day 8 im-scWAT. *P*-values by two-tailed Mann–Whitney U test. **n** H3K9me2 ChIP-qPCR in day 8 im-scWAT transfected with si-Ctrl, si-*Mypt1* #2 + si-*Ppp1cb* #1. **o** MYPT1-PP1β dephosphorylates pS265-JMJD1A and suppresses beige adipogenesis via epigenetic rewriting. (**d–f, i–l, n**) Representative of three (**e–f, i–l**) or two (**d, n**) independent experiments. Data are mean ± SEM of three technical replicates in (**i–k, n**). Source data are provided as a Source data file.

phosphorylation, we transduced retroviral vectors expressing WT or a phospho-defective mutant human RLC (Thr18-to-Ala/Ser19-to-Ala (TA/SA)-human RLC) into im-scWAT cells (Fig. 2g, Supplementary Fig. 2e). Consistent with our model, expression of phospho-defective RLC markedly reduced thermogenic gene induction by MYPT1 depletion (Fig. 2h, Supplementary Fig. 2f). In addition, inhibition of actomyosin tension during beige adipogenesis using blebbistatin, a myosin II inhibitor[20], profoundly reduced thermogenic gene induction by MYPT1 depletion (Fig. 2i, Supplementary Fig. 2g). Collectively, these results identify the phospho-RLC-mediated signaling pathway as a crucial pathway for *Ucp1* regulation by MYPT1-PP1β (Fig. 2j).

## MYPT1 orchestrates epigenetic and transcriptional pathways

Taken together, our data suggest that phospho-JMJD1A-mediated histone demethylation pathway and phospho-RLC induced actomyosin tension and its downstream signaling pathway coordinately promote beige adipogenesis (Fig. 3a). To evaluate this at the molecular level, we performed RNA-seq in MYPT1-depleted im-scWAT cells during beige adipogenesis (day 3) to investigate pathways downstream of MYPT1-PP1β. KEGG pathway analysis of genes that were increased following MYPT1 depletion in the RNA-seq data (Supplementary Fig. 3a) set revealed that these genes were associated with the Hippo signaling pathway (Fig. 3b). In addition, motifs for the transcription factor of (TEAD) were identified in the promoter region of these genes (Fig. 3c). The YAP/TAZ transcriptional co-activators are key downstream effectors of Hippo signaling pathway[21]. They undergo regulated shuttling between the cytoplasm and nucleus and interact with TEAD transcription factors to activate gene expression[14]. Indeed, qPCR analysis revealed that several known YAP/TAZ target genes (*Ctgf* and *Cyr61*) were upregulated by MYPT1 depletion (Supplementary Fig. 3b), indicating that inactivation of MYPT1-PP1β enhances YAP/TAZ transcriptional activity. It is known that YAP/TAZ activity is regulated by actomyosin tension, thereby mediating the nuclear transduction of mechanical and cytoskeletal signaling pathways[13,15]. A previous report showed that actomyosin tension regulates *Ucp1* expression through YAP/TAZ recruitment to the *Ucp1* enhancer[16] and consistent with this report, we showed that blebbistatin, a myosin II inhibitor that disrupts actomyosin-dependent regulation through YAP/TAZ, decreases MYPT1-PP1β mediated regulation of *Ucp1* (Fig. 2i). Therefore, we asked whether YAP/TAZ signaling might be involved in *Ucp1* regulation by MYPT1-PP1β in beige adipocytes. Since *Taz* expression was much higher than *Yap* in im-scWAT cells and it increased during beige adipogenesis (Supplementary Fig. 3c), we knocked down TAZ to assess the role of YAP/TAZ signaling. TAZ depletion reduced the expression of *Ucp1* (Supplementary Fig. 3d). Importantly, thermogenic gene induction by MYPT1 depletion was abrogated by TAZ depletion (Fig. 3d, Supplementary Fig. 3e), indicating that MYPT1-PP1β requires the YAP/TAZ signaling for *Ucp1* regulation. Next, to explore the crosstalk between H3K9me2 demethylation and

YAP/TAZ activation, we analyzed JMJD1A ChIP-seq data and found that JMJD1A binding on day 4 of beige adipogenesis is enriched near genes from Fig. 3b that were increased following MYPT1 depletion (Fig. 3e). In addition, the location of JMJD1A binding in these upregulated genes were significantly closer to the gene transcription start site (TSS) compared to unaffected genes (Supplementary Fig. 3f). Importantly, TEAD motifs were also enriched in the JMJD1A peaks annotated to genes that were upregulated following MYPT1 depletion (Fig. 3f). However, the TEAD motifs were not found in JMJD1A peaks associated with genes that were downregulated or unaffected by MYPT1 depletion (Supplementary Fig. 3g). We also found that a JMJD1A peak localized to *Ucp1* enhancer regions contains the TEAD motifs (Fig. 3g). These observations prompted us to hypothesize that phospho-JMJD1A-mediated H3K9me2 demethylation is a prerequisite for YAP/TAZ-mediated transcriptional activation of *Ucp1*. To investigate this, we introduced *WT*- or the demethylation defective H1120Y-human JMJD1A mutant into im-scWAT cells and showed that the increased expression of *Ucp1* following si-*Mypt1* was completely blocked following H1120Y-JMJD1A transduction (Fig. 3h). The induction of other thermogenic genes was also inhibited by the transduction of H1120Y-JMJD1A (Supplementary Fig. 3h). We also showed that thermogenic gene expression in im-scWAT pre-adipocytes transduced with H1120Y-JMJD1A was rescued by overexpression of WT-JMJD1A (Fig. 3i, Supplementary Fig. 3i). Furthermore, in im-scWAT overexpressing WT-human JMJD1A, H3K9me2 levels in the *Ucp1* enhancer were reduced by *Mypt1* knockdown (Fig. 3j). In contrast, in H1120Y-human JMJD1A-overexpressing im-scWAT, H3K9me2 levels on the *Ucp1* enhancer were high, and *Mypt1* knockdown hardly reduced H3K9me2 levels (Fig. 3j). These studies strongly suggest that JMJD1A-mediated H3K9me2 is pivotal for the subsequent YAP/TAZ-mediated transcriptional activation of *Ucp1* (Fig. 3k).

## β-AR-PKA signaling inhibits MYPT1-PP1β phosphatase activity

MYPT1 has consensus sites for protein kinases and its phosphorylation is important for MYPT1-PP1β activity[22–24]. To understand how MYPT1-PP1β activity might be regulated by cold stress/β-AR-PKA signaling, phosphopeptides from im-scWAT cells treated with ISO were analyzed by LC-MS/MS (Fig. 4a). MYPT1 derived peptides that were phosphorylated at T694 (in murine) were detected and its phosphorylation level was significantly increased by ISO stimulation (Fig. 4a–c). T694 of MYPT1 is located within a putative consensus sequence for PKA phosphorylation (Fig. 4d). ISO-induced T694-MYPT1-phosphorylation was confirmed by immunoblotting with a phospho antibody prepared against this motif (Fig. 4e). Moreover, treatment of ISO stimulated cells with the selective PKA inhibitor H89 reduced MYPT1 phosphorylation (Fig. 4f). To investigate the effect of MYPT1 phosphorylation on MYPT1-PP1β activity, we transduced retroviral vectors expressing a WT or phospho-defective (Thr696 in human -to-Ala) mutant human MYPT1 into im-scWAT cells (Supplementary Fig. 4a, b). We simultaneously

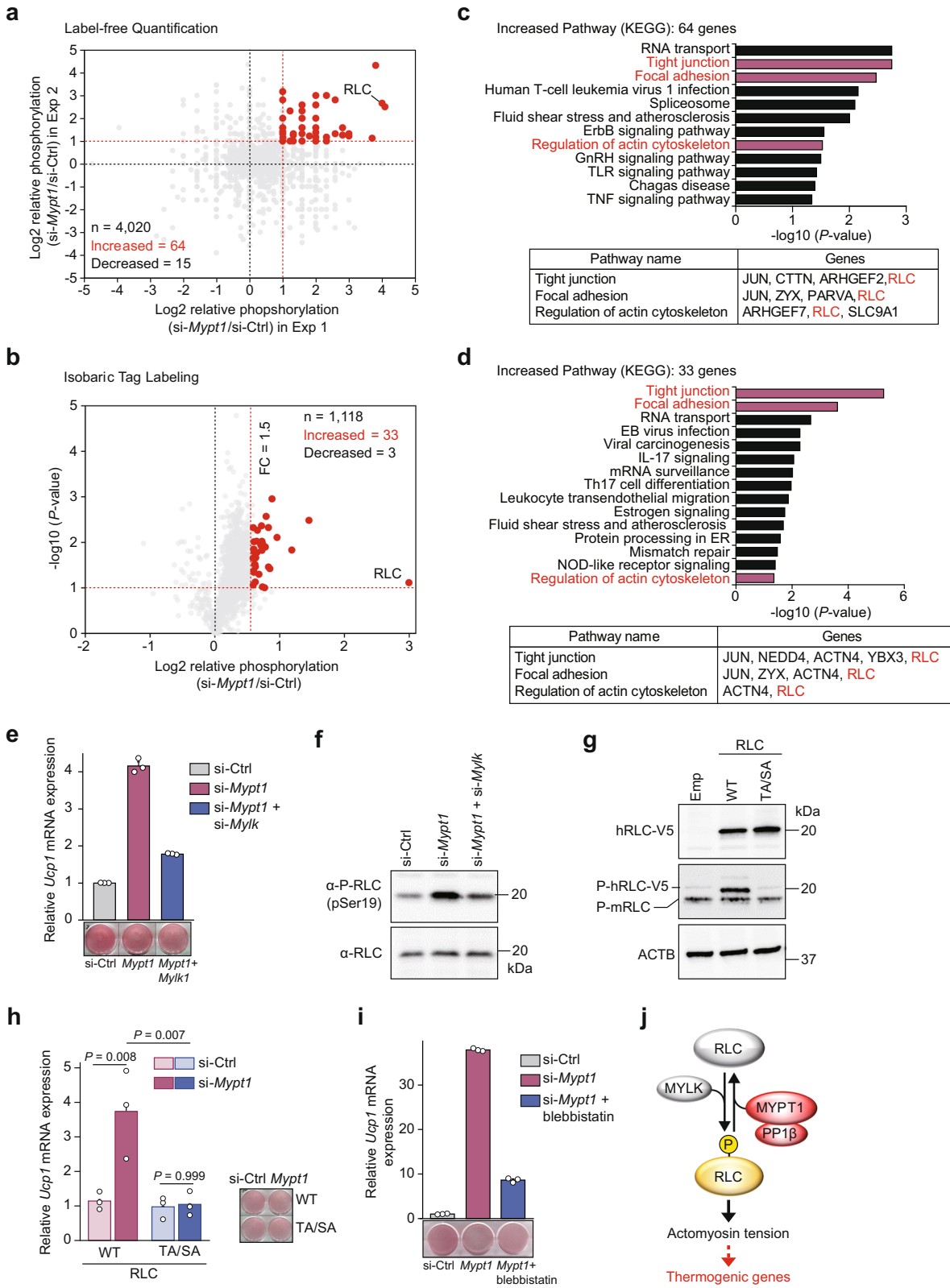

depleted these cells of endogenous murine MYPT1 using siRNA specifically designed to murine RNA sequence of MYPT1 (si-*Mypt1* #3, Supplementary Fig. 4c). Compared to WT-MYPT1 cells, ISO-induced JMJD1A phosphorylation was markedly decreased in T696A-MYPT1 cells (Fig. 4g), indicating that pT696-MYPT1 is inhibitory to MYPT1-PP1β activity. Consistent with changes in phosphorylation levels, thermogenic gene mRNAs were much lower in T696A-MYPT1 expressing cells

than in WT-MYPT1 expressing cells (Fig. 4h). Although previous studies reported mono-phosphorylation at a neighboring S695 (in human) or di-phosphorylation (S695 and T696) of MYPT1 by PKA in vitro and in smooth muscle[22,23,25], our data indicated that the primary phosphorylation site of MYPT1 under β-AR activation in scWAT cells is T694 and this phosphorylation inhibits MYPT1-PP1β activity. Together, these results suggest that β-AR activation inhibits MYPT1-PP1β activity via

**Fig. 2 | MYPT1-PP1β targets RLC phosphorylation for beige adipogenesis.**
**a** Scatter plot of si-*Mypt1* #2/si-Ctrl phosphoproteome in day 0 im-scWAT (Label-free Quantification) (Experiments 1 and 2). *X*-axis and *Y*-axis, log2 FC of total spectrum count in Experiments 1 and 2, respectively. Number of total phospho-peptides (n), phosphopeptides with increased (Increased: FC > 2) and decreased (Decreased: FC < 0.5) phosphorylation levels by MYPT1 depletion. **b** Volcano plot of si-*Mypt1* #2/si-Ctrl phosphoproteome in day 0 im-scWAT (Isobaric Tag Labeling) (the same samples were subjected to LC-MS/MS for three times as technical replicates to verify measurement accuracy). *X*-axis, log2 FC of reporter ion inten-sity: *Y*-axis, −log10 *P*-value of reporter ion intensity change. The number of total phosphopeptides (**n**), phosphopeptides with increased (Increased: FC > 1.5, *P*-value < 0.1) and decreased (Decreased: FC < 1/1.5, *P*-value < 0.1) phosphorylation levels by MYPT1 depletion. **c**, **d** KEGG pathway analysis of 64 and 33 peptides phosphorylated by MYPT1 depletion in (**a**) and (**b**), respectively (upper).

Enrichment of pathways including RLC (bottom). **e** *Ucp1* mRNA levels in day 8 im-scWAT transfected with si-Ctrl, si-Ctrl + si-*Mypt1* #2, or si-*Mypt1* #2 + si-*Mylk* (upper). ORO staining on day 8 (bottom). **f** P-RLC and RLC immunoblotting from day 0 im-scWAT transfected with si-Ctrl, si-*Mypt1* #2 + si-Ctrl, or si-*Mypt1* #2 + si-*Mylk*. **g** V5, P-RLC, or ACTB immunoblotting from WT- or T18A/S19A-human RLC-transduced im-scWAT on day 0. **h** *Ucp1* mRNA levels in WT- or T18A/S19A-human RLC-transduced im-scWAT on day 8 transfected with si-Ctrl or si-*Mypt1*#2 (left). Data are mean ± SEM of three biological replicates. One-way ANOVA with Tukey's multiple com-parisons test. ORO staining on day 8 (right). **i** *Ucp1* mRNA levels in day 8 im-scWAT treated with si-Ctrl, si-*Mypt1*#2, or si-*Mypt1* #2 + blebbistatin (top). ORO staining on day 8 (bottom). **j** RLC phosphorylation and resultant actomyosin tension is pivotal for MYPT1-mediated *Ucp1* regulation. **e**, **f**, **g**, **i** Representative of three (**e**, **i**) or two (**f**, **g**) independent experiments. Data are mean ± SEM of three technical replicates in (**e**) and (**i**). Source data are provided as a Source data file.

PKA-mediated MYPT1 phosphorylation at T694, which induces phos-phorylation of JMJD1A to facilitate beige adipogenesis (Fig. 4i).

## MYPT1 is crucial for scWAT beiging in mice
To investigate the role of MYPT1 in beiging in whole animals, a control adeno-associated virus (AAV) (AAV-CMV-mCherry) or Cre recombi-nase AAV (AAV-CMV-mCherry-2A-Cre) was injected into inguinal scWAT of *Mypt1^flox/flox* mice[26] (Fig. 5a, Supplementary Fig. 5a–c). Expression of mCherry-2A-Cre was confirmed by immunohistochem-istry using anti-mCherry antibody (Supplementary Fig. 5d). These mice were acclimated at room temperature (RT) and then housed at 12 °C for 1 week. Injection of AAV expressing Cre significantly increased UCP1 protein levels (Fig. 5b), thermogenic gene mRNA expression (Fig. 5c, left) and NE-induced OCR (Fig. 5c, right) in scWAT. Histological analysis showed that, in the scWAT of *Mypt1^flox/flox* mice injected with AAV-mCherry-2A-Cre, expression of UCP1 was induced in multilocular lipid droplets containing adipocytes, and many of the UCP1-positive cells co-expressed mCherry-2A-Cre, suggesting that MYPT1 deficiency induced the production beige adipocytes in mice (Fig. 5d and Sup-plementary Fig. 5e). Injection of AAV-mCherry induces mCherry expression in white adipocytes, but this by itself does not recruit beige adipocytes (Supplementary Fig. 5f). Reduced MYPT1 protein by AAV-mCherry-2A-Cre injection slightly decreased scWAT weight without affecting body weight (Fig. 5e, f). Similarly, higher *Ucp1* levels and OCR were observed in Cre recombinase adenovirus infected primary scWAT cultures of *Mypt1^flox/flox* mice compared to control cells (Sup-plementary Fig. 5g, h). These results provide evidence that MYPT1 depletion promotes beige adipogenesis in vivo.

Next, *Mypt1^flox/flox* mice were crossed with *Adipoq*-Cre mice to delete *Mypt1* gene in adipocytes (Supplementary Fig. 5i)[27]. Although *Mypt1* levels were reduced in scWAT compared with those in control mice, *Mypt1^flox/flox::Adipoq*-Cre mice were not more cold tolerant (Sup-plementary Fig. 5j, k). In addition, *Ucp1* levels in adipose tissues of these mice housed at 12 °C for 1 week were indistinguishable from those of control mice (Supplementary Fig. 5k). Body weight changes during acute cold exposure and weights of adipose tissues depots of these mice housed at 12 °C for 1 week were similar to control mice (Supplementary Fig. 5l).

To further investigate the role of MYPT1 in de novo recruitment of beige adipocytes, we depleted *Mypt1* in pre-adipocytes for brown and white adipocytes by crossing *Mypt1^flox/flox* mice with *Pdgfra*-Cre mice[28–30]. Because *Mypt1^flox/flox::Pdgfra*-Cre mice were embryonic lethal as reported in conventional *Mypt1* knockout mice[31], we analyzed heterozygous *Mypt1^+/flox::Pdgfra*-Cre mice. In scWAT culture of *Mypt1^+/flox::Pdgfra*-Cre mice, *Mypt1* levels were reduced by 50% (note that these mice are het-erozygote) and thermogenic genes were increased compared to those in control cells (Supplementary Fig. 5m). When *Mypt1^+/flox::Pdgfra*-Cre mice were housed at 12 °C for 1 week, UCP1 protein levels in the scWAT were significantly higher than those in control mice (Fig. 5g). Phos-phorylation of JMJD1A was higher in the scWAT of *Mypt1^+/flox::Pdgfra*-Cre

mice than in that of control mice (Fig. 5g, inset). Expression of ther-mogenic genes was also increased in the scWAT of *Mypt1^+/flox::Pdgfra*-Cre mice, even in RT housing (Fig. 5h). Histological analysis showed that *Mypt1^+/flox::Pdgfra*-Cre mice had more UCP1-positive multilocular beige adipocytes in scWAT than did control mice acclimated at RT (Fig. 5i). The scWAT weights of the mice did not differ between the two groups (Supplementary Fig. 5n). In contrast, *Ucp1* expression and tissue weight did not change in BAT between the groups (Supplementary Fig. 5o, p). *Mypt1^+/flox::Pdgfra*-Cre mice were cold tolerant (Supplementary Fig. 5q). In addition, HFD feeding at RT resulted in higher body weight gain in control mice, while *Mypt1^+/flox::Pdgfra*-Cre mice exhibited reduced body weight gain in comparison (Fig. 5j). Furthermore, *Mypt1^+/flox::Pdgfra*-Cre mice showed improved glucose tolerance and lower serum insulin levels under fasting conditions or after glucose injection compared with control mice (Fig. 5k, l, Supplementary Fig. 5r), indicating that MYPT1 depletion is associated with a higher energy consumption phenotype and improved glucose metabolism. These results indicate that MYPT1 is a crucial regulator of thermogenic capacity in scWAT and energy metabolism in vivo.

## Discussion
Spatiotemporal gene expression is essential for cell fate decision and cellular responses to environmental cues including cold stress, which are regulated by a combination of epigenetic and direct transcriptional regulatory events. Mammals cope with chronic cold stress through metabolic adaptation referred to as "beiging" of subcutaneous white adipose tissue, which involves the recruitment of thermogenic adipocytes called beige adipocytes that generate heat through *Ucp1*-driven uncoupled respiration. However, the mechanism for how β-AR-PKA signaling coordinates the action of epigenetic and DNA sequence-dependent transcriptional regulatory complexes to induce beige adipogenesis upon cold stress is incompletely understood.

In this study, we uncovered a role for the MYPT1-PP1β phosphatase complex in the regulation of both chromatin dynamics and co-activator recruitment during beige adipogenesis. We show that the regulatory subunit MYPT1 is phosphorylated at T694 by PKA in response to β-AR signaling, which leads to the inhibition of MYPT1-PP1β activity, thus enabling the efficient shift of chromatin landscape and transcriptional program required for beige adipogenesis in vivo. Mechanistically, we show that MYPT1-PP1β dephosphorylates two substrates: JMJD1A and RLC. Phosphorylation of these substrates upon activation of β-AR sig-naling facilitates the chromatin opening of *Ucp1* gene through JMJD1A-mediated H3K9 demethylation and the recruitment of transcriptional co-activators YAP/TAZ through RLC-mediated actomyosin tension (Fig. 6). Thus, our study reveals a mechanism for how β-AR signaling is integrated into chromatin landscape and transcriptional program that regulate expression of thermogenic genes in response to cold stress.

Phosphorylation of S265 of JMJD1A stimulates its H3K9me2 demethylase activity which is for thermogenic gene regulation in beige adipogenesis[1]. We show that inhibition of MYPT1-PP1β prolongs the life

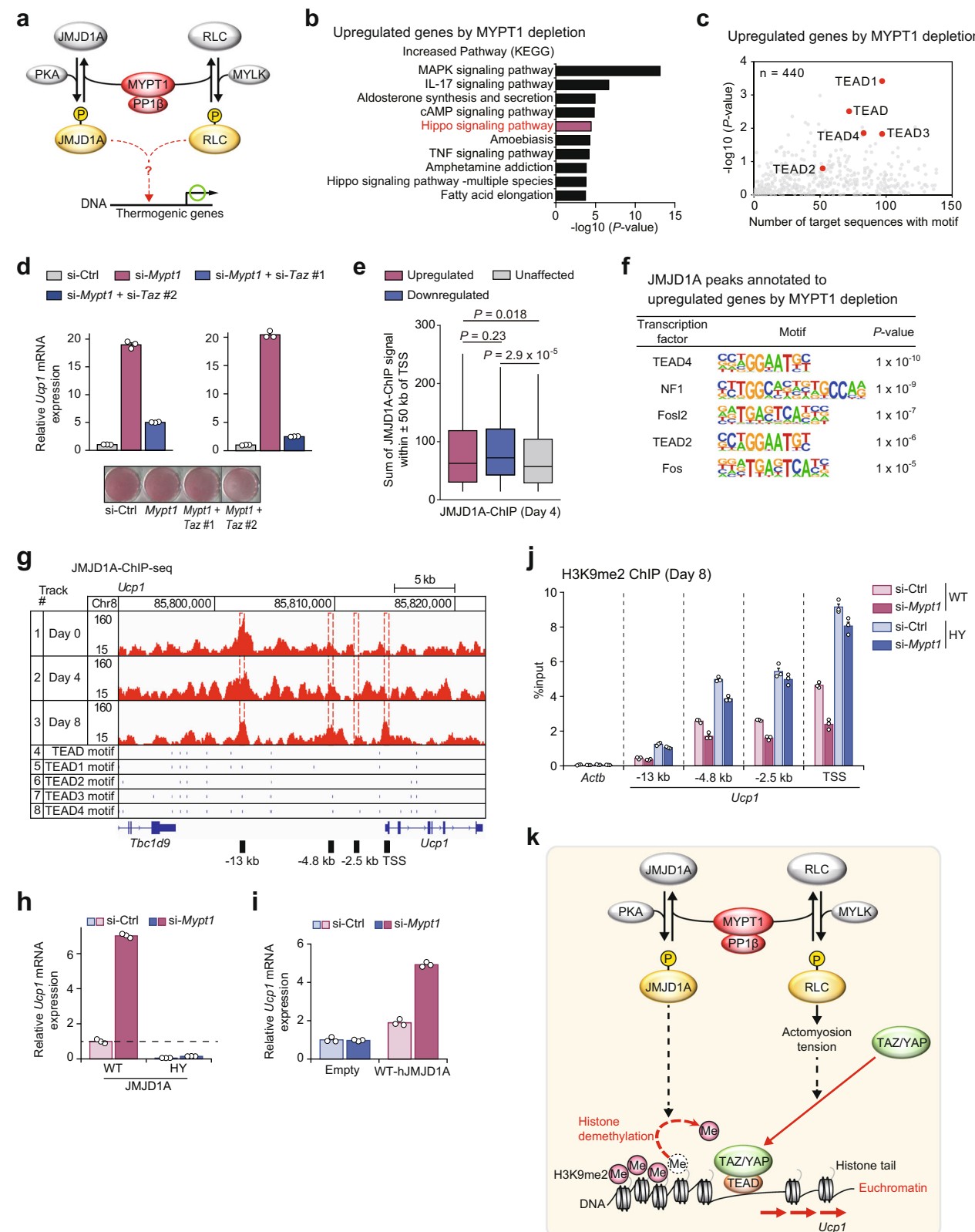

of phosphorylated JMJD1A which increases H3K9me2 demethylation on thermogenic genes, leading to enhanced thermogenic capacity of beige adipocytes.

We also showed that MYPT1-PP1β regulates the RLC-actomyosin-YAP/TAZ pathway for activating *Ucp1* transcription. Although a recent study showed that actomyosin tension regulates thermogenic gene transcriptions via YAP/TAZ[16], the regulatory mechanism was elusive. We found that *Ucp1* induction by MYPT1-PP1β inactivation was blocked by expressing demethylation defective JMJD1A, indicating that JMJD1A-meditaed demethylation is a prerequisite for recruiting YAP/TAZ to *Ucp1* promoter chromatin for robust gene activation. Therefore, MYPT1-PP1β orchestrates the chromatin status and transcriptional programs for beige adipogenesis under cold stress/β-AR signaling.

**Fig. 3 | MYPT1-PP1β orchestrates epigenetic and transcriptional pathways for beige adipogenesis. a** How JMJD1A-dependent and RLC-dependent pathway can be integrated to regulate beige adipogenesis remains elusive. **b** KEGG pathway analysis of top 200 genes upregulated by MYPT1 depletion on day 3 im-scWAT differentiated for beige adipogenesis. **c** Motifs identified in 2 kb upstream region of TSS of upregulated genes by MYPT1 depletion in (**b**). Y-axis, −log10 (P-value): X-axis, the number of sequences containing that motif. **d** *Ucp1* mRNA levels on day 8 in im-scWAT transfected with si-Ctrl, si-*Mypt1* #2 + si-Ctrl, or si-*Mypt1* #2 + si-*Taz* (left: si-*Taz* #1, right: #2). ORO staining on day 8 (bottom). **e** Box plot showing JMJD1A binding signal near unaffected, upregulated (FC > 1.5, RPKM of si-*Mypt1* > 1) and downregulated (FC < 1/1.5, RPKM of si-Ctrl > 1) genes by MYPT1 depletion in day 3 im-scWAT (*n* = 11,681, *n* = 424, and *n* = 229, respectively). The box shows the median, and first and third quartiles. Genome-wide analyses were performed once on the basis of JMJD1A ChIP-seq data set in day 4 differentiated im-scWAT. *P*-values by two-tailed Mann–Whitney test. **f** Motifs identified in JMJD1A peaks on day 4 annotated to upregulated genes by MYPT1 depletion in (**e**) by known motif searching. **g** Genome browser views showing ChIP-seq profiles for JMJD1A on *Ucp1* genomic regions in im-scWAT on day0, 4, and 8 and TEAD motifs located in the regions. **h** *Ucp1* mRNA levels in WT- or H1120Y-human JMJD1A-transduced im-scWAT on day 8 transfected with si-Ctrl or si-*Mypt1* #2. **i** *Ucp1* mRNA levels in empty vector or WT-human JMJD1A-transduced im-scWAT expressing H1120Y-human JMJD1A on day 8 transfected with si-Ctrl or si-*Mypt1* #2. **j** H3K9me2 ChIP-qPCR in day 8 im-scWAT overexpressing WT- or H1120Y- hJMJD1A transfected with si-Ctrl) or si-*Mypt1*#2. **k** MYPT1-PP1β targets phosphorylation of JMJD1A for chromatin accessibility and RLC for YAP/TAZ-mediated transcription to suppress beige adipogenesis. (**d**, **h**–**j**) Representative of three (**h**) or two (**d**, **i**, **j**) independent experiments. Data are mean ± SEM of three technical replicates in (**d**, **h**–**j**). Source data are provided as a Source data file.

The overall mechanism is driven by direct phosphorylation and inactivation of MYPT1-PP1β activity during cold stress through β-AR signaling activation of PKA. This provides a rapid 'On-Off' phospho-switch in response to environmental temperature changes. PKA is rapidly activated by β-AR signaling leading to inactive MYPT1-PP1β through PKA-mediated MYPT1 phosphorylation. The consequent PKA activation and MYPT1-PP1β inactivation leads to high levels of active phosphorylated JMJD1A. This results in robust activation of *Ucp1* expression and uncoupled respiration. When β-AR signaling is decreased as the cold stress is reversed (e.g., warm environment) PKA is de-activated favoring active unphosphorylated MYPT1-PP1β, which rapidly turns off the thermogenic program by inactivating JMJD1A through de-phosphorylation. This phosphorylation/de-phosphorylating cycle orchestrated by MYPT1-PP1β is a well-organized physiological system for rapidly coping with cold stress.

Besides cold-induced sympathetic nerve activation, humoral factors contribute to beige adipogenesis through inter-organ crosstalk[32–34]. We found that *Mypt1* gene deletion in scWAT through AAV-mediated Cre recombination increased *Ucp1* expression, indicating that MYPT1-mediated thermoregulation is attributable to the tissue-autonomous function of MYPT1 independent of higher order levels of endocrine or neuronal system controls. Cell culture experiments indicated that *Mypt1* depletion in pre-adipocytes was sufficient for the induction of the beige adipogenic program. We generated both pre-adipocyte- and adipocyte-specific *Mypt1*-deficient mice but only pre-adipocyte-specific *Mypt1*-depletion increased *Ucp1* expression in scWAT, suggesting that MYPT1 contributes to de novo differentiation of beige adipocytes but is probably not involved in *trans*-differentiation of mature white adipocytes. *Mypt1* deficiency did not influence *Ucp1* levels by chronic cold stress in BAT, indicating that MYPT1 may not be essential for BAT thermogenic function at least during the chronic phase of cold adaptation. However, it is also possible that the 50% reduction in MYPT1 was sufficient to reveal a phenotype due to reduction in scWAT but not in BAT where basal levels of *Ucp1* are already very high. Collectively, our results indicate that MYPT1 is a key negative regulator of de novo differentiation of beige adipocytes under physiologically relevant conditions.

Overall, the present study reveals a previously undescribed mechanism of MYPT1-PP1β phosphatase regulation of two intersecting pathways that together regulate beige adipogenesis downstream of β-AR signaling. This represents a key connection between epigenetic and direct transcriptional mechanisms in the control of thermogenesis and suggests MYPT1 and its interacting proteins could be molecular targets for inducing beige adipogenesis for both natural and therapeutic situations where increased thermogenesis is beneficial.

## Methods
A list of reagents and resources used is provided in Supplementary Table 2.

## Antibodies
Mouse monoclonal antibodies, IgG-F0618 and IgG-F0231, against mouse JMJD1A (amino acids 843–893) and human monoclonal antibody, IgG-F3640, against human JMJD1A (amino acids 841-891) were produced by immunizing mice with gp64 fusion protein-expressing baculovirus as described[9]. A rabbit polyclonal phospho-specific antibody against JMJD1A S265 was produced with a synthetic phosphopeptide corresponding to residues surrounding S265 JMJD1A (Ac-C-KRKS(pS)ENNGS-amide) as described previously[9]. Information on antibody dilutions/amounts, validation, company names, catalog numbers and clone numbers for monoclonals in the Reporting Summary and Methods section for all assays using antibodies. Information on all the antibodies used in this study are listed in Supplementary Table 2.

## Primary adipocyte culture from subcutaneous white adipose tissue (scWAT)
SVF of inguinal scWAT was obtained from 2-week-old mice as described previously[6,35]. scWATs were first washed in phosphate-buffered saline (PBS) before being subjected to enzymatic digestion with 1.5 mg/mL collagenase (Wako) and 2.4 U/mL dispase II (Roche) for 40 min at 37 °C to obtain a single-cell suspension. After digestion, the centrifuged cell pellet, termed SVF, was resuspended in the complete medium (Dulbecco's modified Eagle's medium/F12 (DMEM/F12) (Sigma-Aldrich), containing 1.2 g/L NaHCO3 and 10% fetal bovine serum before serial filtration through a 100 μm nylon cell strainer (BD Falcon). To remove the red blood cells, immune cells, and other contaminants, SVF were plated on collagen-coated plates, and 2 h later, the medium was aspirated, washed with PBS four times, and fresh medium was added.

## Adipocyte differentiation
The induction of beige adipocytes from inguinal scWAT was performed as described previously[6,35] with the following modifications. Briefly, primary pre-adipocytes cultured to -100% confluence in complete medium were induced for differentiation with induction medium (i.e., complete medium containing 0.125 mM indomethacin, 5 μM dexamethasone, 0.5 mM 3-isobutyl-1-methylxanthine, 0.5 μM Rosiglitazone (Rosi), 5 μg/mL insulin, and 1 nM T3) (day 0). After 2 days, the cells were incubated in the maintenance medium with 0.5 μM Rosi (i.e., complete medium containing only insulin, T3, and Rosi) for another 6 days before the assay (day 8). im-scWAT pre-adipocytes were derived from inguinal scWATs established by infecting isolated SVFs of inguinal WAT with retrovirus-expressing large T antigen pBabe SV40 Large T antigen from Addgene (no. 13970)[6,35]. For the induction of beige adipocytes, im-scWAT pre-adipocytes were cultured in DMEM supplemented with 10% FBS at 37 °C in 5% $CO_2$ until confluent (day 0), and thereafter treated with the culture medium containing 0.125 mM indomethacin, 5 μM dexamethasone, 0.5 mM 3-isobutyl-1-methylxanthine, 0.5 μM Rosi, 5 μg/mL insulin, and 1 nM T3 for 48 h (day 2), followed by treatment with only insulin, T3, and Rosi for another 6 days, with medium replacement every 2 days (day 8)[6,35].

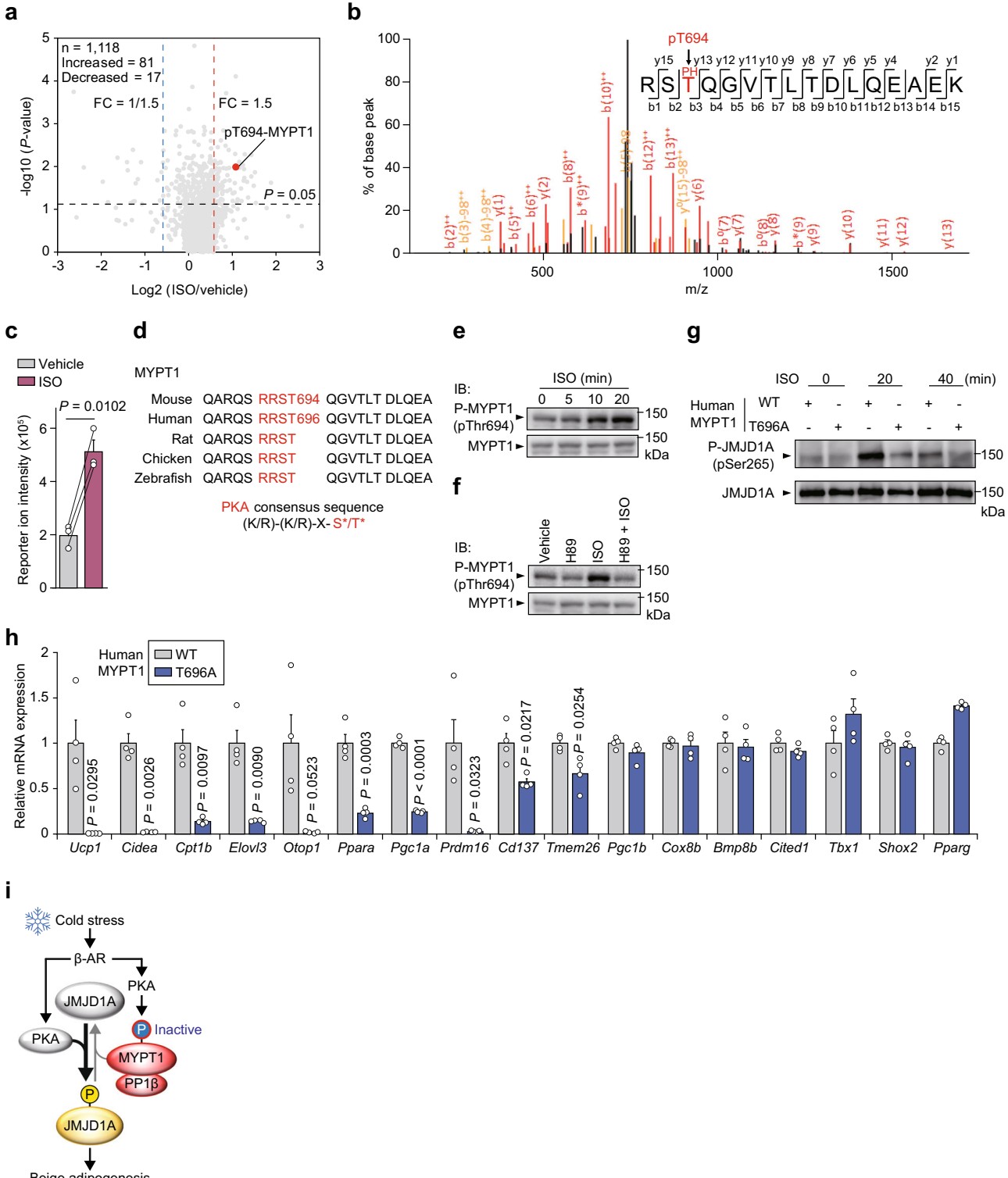

**Nature Communications** | (2022)13:5715

## Other cell culture

3T3-L1 pre-adipocytes (ACTT, Cat#ATCC CL-173), NIH-3T3 cells (ATCC, Cat#ATCC CRL-1658), Plat E packaging cells (Cosmo Bio, Cat#RV-101), and AAV-293 cells were maintained in basal medium.

## AR activation, protein phosphatase inhibition, and myosin II inhibition

To activate the non-selective β-AR agonist, isoproterenol (ISO, 1–3 µM for 1 to 8 h) was added to the culture medium. To inhibit PP1 and PP2A, 4 nM calyculin A was added to the culture medium for 30 min. To inhibit actomyosin tension, the cells were differentiated for beige adipogenesis in media supplemented with 10 µM blebbistatin.

## Oil red O (ORO) staining

Cells at the specified stages of differentiation were rinsed with PBS and fixed with 3.7% formaldehyde in $H_2O$ for 10 min. After two washes in PBS and one wash with 60% isopropanol for 1 min, cells were stained for 15 min in freshly diluted ORO solution (0.18% (w/v) ORO in 60% isopropanol). The stain was then removed, and the cells were washed

**Fig. 4 | β-AR-PKA phosphorylation of MYPT1 inhibits MYPT1-PP1β activity.**
**a** Volcano plot of ISO/Vehicle phosphoproteome in im-scWAT pre-treated with ISO (10 μM) for 40 min on day 0 (Isobaric Tag Labeling) (the same samples were processed and subjected to LC-MS/MS analysis for three times as technical replicates to verify measurement accuracy). X-axis, log2 FC of reporter ion intensity: Y-axis, −log10 P-value of reporter ion intensity change. The number of total phosphopeptides (n) and phosphopeptides with increased (FC > 1.5, P < 0.05) and decreased (FC < 1/1.5, P < 0.05) phosphorylation levels by MYPT1 depletion are shown. **b** Post-translational modifications of MYPT1 identified by phosphoproteomics in (a). MS/MS spectrum of P-MYPT1 fragment from R692 to K707. **c** TMT reporter ion intensities derived from MS/MS spectrum of P-MYPT1 fragment. Data are mean ± SEM of three technical replicates. P-values by paired two-tailed t test. **d** PKA consensus site is conserved among various species. **e** P-MYPT1 and MYPT1 immunoblotting from im-scWAT pre-cultured in 0.1% bovine serum albumin containing

DMEM for 3 h and treated with ISO (10 μM) for 0, 5, 10, and 20 min on day 0 (**f**) P-MYPT1 and MYPT1 immunoblotting from im-scWAT pre-cultured in 0.1% bovine serum albumin containing DMEM for 3 h and treated with H89 (20 μM) for 20 min and then treated with ISO (10 μM) for 10 min on day 0. **g** P-JMJD1A and JMJD1A immunoblotting following immunoprecipitation with anti-mouse JMJD1A from WT- or T696A-human MYPT1-transduced im-scWAT treated with ISO (10 μM) for 0, 20, and 40 min on day 0 transfected with si-*Mypt1* #3. **h** Thermogenic gene mRNA expression in day 8 im-scWAT differentiated for beige adipogenesis transduced with WT- or T696A-human-MYPT1 transfected with si-*Mypt1* #3. Data are mean ± SEM of four biological replicates. P-values by two-tailed Student's t test. **i** PKA is activated upon β-AR activation, while MYPT1-PP1β is inactivated via Thr694 phosphorylation of MYPT1 by PKA, enabling efficient phosphorylation of JMJD1A in response to cold stress. **e, f, g** Representative of three independent experiments. Source data are provided as a Source data file.

## Immunoprecipitation and immunoblotting analyses
Whole cell lysates (WCLs) were collected in cell lysis buffer A (50 mM Tris-HCl, pH 8.0, 0.1 mM ethylenediaminetetraacetic acid (EDTA), 5% glycerol, 100 mM KCl, 0.1% NP-40) containing protease inhibitor cocktail (Nacalai Tesque), 1 mM phenylmethylsulfonyl fluoride, and phosphatase inhibitors (40 mM NaF and 1 mM $Na_3VO_4$). JMJD1A protein was immunoprecipitated in cell lysis buffer A for 4 h at 4 °C using the antibodies described in the legend to figures. Co-immunoprecipitates were subjected to sodium dodecyl sulfate-polyacrylamide gel electrophoresis (SDS-PAGE) and the immunoblots were visualized by chemiluminescence using Super Signal West Dura Extended Duration Substrate (Thermo Fisher Scientific). Luminescent images were analyzed using ImageQuant LAS 4000 mini (GE Healthcare) as described previously[9,36] For quantification of immunoblot, ImageJ1.53k software was used. Primary antibodies used: anti-mouse JMJD1A mouse mAb IgG-F0618 (RCAST, The University of Tokyo, Japan, 10 μg/mL for IB, 8.3 μg/mL for IP), anti-mouse JMJD1A mouse mAb IgG-F0231 (RCAST, The University of Tokyo, Japan, 4 μg/mL for IB, 12 μg/mL for IP), anti-human JMJD1A mouse mAb IgG-F3640 (RCAST, The University of Tokyo, Japan, 10 μg/mL for IP), anti-mouse P-JMJD1A (pS265) rabbit pAb 11890-2 (RCAST, The University of Tokyo, Japan, 1:500 for IB), anti-RLC rabbit mAb D18E2 (Cell Signaling Technology, 8505, 1:500 for IB), anti-P-RLC (pSer19) rabbit pAb (Cell Signaling Technology, 3671, 1:500 for IB), anti-MYPT1 rabbit pAb (Cell Signaling Technology, 2634, 1:500 for IB), anti-P-MYPT1 (pThr696) rabbit pAb (Sigma-Aldrich, ABS45, 1:500 for IB), anti-ACTB mouse mAb AC-15 (Sigma-Aldrich, A5441, 1:5000 for IB), anti-V5 mouse mAb (Thermo Scientific, R960-25, 1 μg/mL for IB), anti-UCP1 mouse mAb 536435 (R&D Systems, MAB6158, 0.5 μg/mL for IB), anti-TOM20 rabbit pAb (Proteintech, 11802-1-AP, 0.8 μg/mL for IB), anti-FLAG mouse mAb M2 (Sigma-Aldrich, F3165, 1 μg/ml for IP). Secondary antibodies used: anti-mouse IgG-HRP (Sigma-Aldrich, A4416, 1:10,000 for IB), anti-rabbit IgG-HRP (Sigma-Aldrich, A0545, 1:10,000 for IB).

## RNA interference
The duplexes of each small interfering RNA (siRNA), targeting murine *Mypt1* mRNA (MSS206918 [si-*Mypt1* #1], MSS206919 [si-*Mypt1* #2]:: Invitrogen, J-063177-05-0002 [si-*Mypt1* #3]:: Dharmacon), *Ppp1ca* (J-040960-05-0002 [si-*Ppp1ca* #1], J-040960-06-0002 [si-*Ppp1ca* #2]:: Dharmacon), *Ppp1cb* (MSS237617 [si-*Ppp1cb* #1], MSS237618 [si-*Ppp1cb* #2]:: Invitrogen), *Ppp1cc* (71267 [si-*Ppp1cc* #1], 71360 [si-*Ppp1cc* #2]:: Invitrogen), *Taz* (MSS251009 [si-*Taz* #1], MSS251010 [si-*Taz* #2]:: Invitrogen), *Lats2* (MSS224891 [si-*Lats2* #1], MSS224893 [si-*Lats2* #2]:: Invitrogen), *Mylk* (MSS200831 [si-*Mylk*]:: Invitrogen), and control siRNA (Invitrogen Stealth RNAi, Negative Control Med GC Duplex, #12935-112:: Invitrogen, ON-TARGETplus Non-targeting Control Pool, #D-001810-10-20:: Dharmacon, Silencer™ Negative Control No.

1 siRNA, #AM4611:: Invitrogen) were transfected. The cells were transfected with 2.5 to 20 nM of siRNA using Lipofectamine RNAi MAX reagent (Invitrogen) according to the manufacturer's instructions, with modifications as follows. Lipofectamine RNAi MAX reagent (Invitrogen)/siRNA complexes were formed in 0.4 mL of Opti-MEM™ I reduced serum medium (Gibco) for 15 min at RT and then added to each well of a 6-well plate (Falcon) in 1.6 mL of complete medium. Cells were cultured for 24–48 h until they reached confluence and were induced for differentiation as described above.

## Plasmid construction and virus preparation
To construct a retroviral vector for JMJD1A (pMXs-FLAG-human JMJD1A-IRES/Puro), we subcloned the DNA sequence encoding the human JMJD1A open reading frame into the LTR promoter-driven expression vector pMX (a kind gift from Dr. Toshio Kitamura, The University of Tokyo) carrying three copies of the FLAG sequence as described previously[9]. For construction of lentiviral vector expressing JMJD1A, human *JMJD1A* open reading frame flanking with V5 tag sequence at 3'end was inserted into CSII-EF-MCS-IRES2-Venus vector, which was kindly gifted from RIKEN bioresource research center (BRC) (RDB04384).To construct a retroviral vector for short hairpin RNA (shRNA) targeting *Jmjd1a*, the forward primer 5-gatctttgccgatgacctttca gataattcaagagagattatctgaaaggtcatcggttttgttcc-3 and reverse primer 5-agctggaacaaaaccgatgacctttcagataatctcttgaattatctgaaaggtcatcggcaaa-3 were annealed and subcloned into the mouse U6 promoter-driven expression vector pRetro/Puro-Super3 (a kind gift from Dr. T. Kitamura), in which puromycin-resistant marker sequences were replaced with neomycin-resistant marker sequences. To construct a retroviral vector for RLC (pMXs-human RLC-V5-IRES/Zeo) or MYPT1 (pMXs-human MYPT1-V5-IRES/Zeo), we subcloned the DNA sequence encoding the human RLC or MYPT1 open reading frame into a pMX expression vector, in which the original puromycin-resistant sequences were replaced by Zeocin™-resistant sequences. Mutant versions of human JMJD1A: H1120Y, human RLC: T19A/S20A, and human MYPT1: T696A were generated by PCR-based site-directed mutagenesis. Retroviruses were produced by transfecting each plasmid into Platinum-E packaging cells (a kind gift from Dr. Toshio Kitamura, The University of Tokyo) using GeneJuice (Novagen). Lentiviruses were generated from 293T cells transfected with each plasmid, pCAG-HIVgp (RDB04394) and pCMV-VSV-G-RSV-Rev (RDB04393) gifted from RIKEN BRC. To establish JMJD1A knockdown 3T3-L1 pre-adipocytes, 3T3-L1 pre-adipocytes were transduced with a retroviral vector expressing shRNA targeting murine *Jmjd1a* under the control of the U6 promoter and were selected by G418. To establish 3T3-L1 pre-adipocytes ectopically expressing FLAG-tagged human JMJD1A, the resultant JMJD1A knockdown 3T3-L1 pre-adipocytes were infected with retrovirus-expressing FLAG-tagged human JMJD1A, and the infected cells were selected using puromycin treatment as described previously[9,36]. To establish im-scWAT cells ectopically expressing V5-tagged human JMJD1A or human RLC, im-scWAT cells were infected with retrovirus-expressing V5-

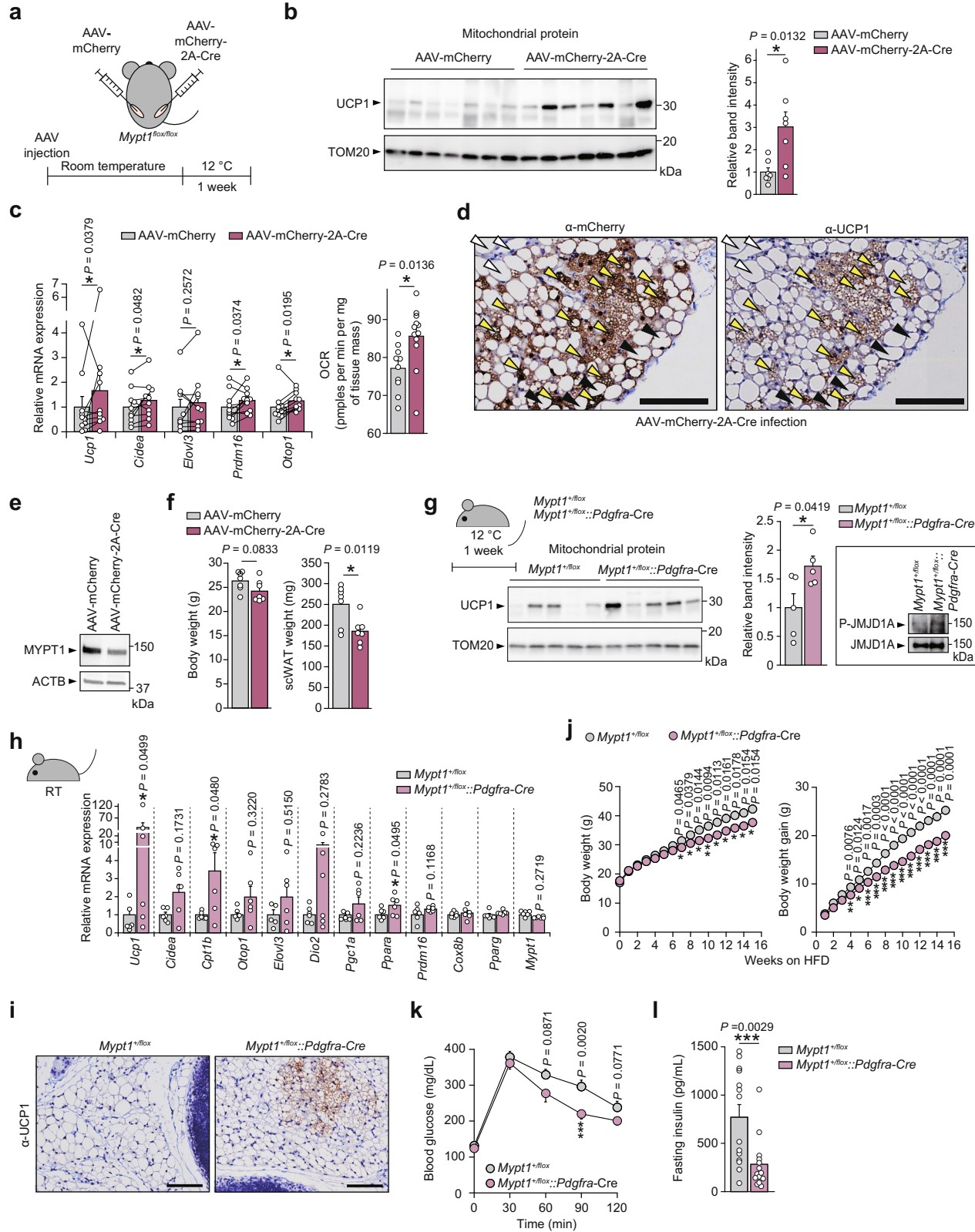

tagged human JMJD1A or human RLC and the infected cells were selected through zeocin treatment (0.1 mg/mL) for 14 days as described previously[9,36]. To establish im-scWAT cells expressing V5-tagged human JMJD1A, the lentiviral-induced Venus-positive cells were isolated by using the FACS Aria II (BD Biosciience). AAV was produced by transfecting AAV plasmids (pAAV-CMV-mCherry or pAAV-CMV-mCherry-2A-Cre, kind gifts from Dr. Akihiro Yamanaka, Nagoya

University), pHelper (Agilent), and pAAV-RC8 (a kind gift from Dr. James M Wilson, The University of Pennsylvania) into AAV-293 cells (Agilent, Cat#240073) using calcium phosphate transfection method.

### Identification of JMJD1A-interacting proteins

3T3-L1 pre-adipocytes retrovirally transduced with an empty vector or with a vector expressing FLAG-tagged human JMJD1A were grown in

**Fig. 5 | MYPT1 is crucial for scWAT beiging in mice. a** *Mypt1^{flox/flox}* mice were injected with AAV-CMV-mCherry or AAV-CMV-mCherry-2A-Cre into inguinal scWAT, and exposed to 12 °C for 1 week. **b** UCP1 immunoblotting of mitochondrial fraction of scWATs from AAV-injected *Mypt1^{flox/flox}* mice (left). Densitometric quantification of UCP1 immunoblotting (right). *n* = 7 per group. **c** Thermogenic gene mRNA levels in scWAT of AAV-injected *Mypt1^{flox/flox}* mice. Data are mean ± SEM (*n* = 10 per group, left). *P*-values by paired two-tailed t-test. NE-induced OCR in scWAT (right). Data are mean ± SEM (AAV-CMV-mCherry, *n* = 10; AAV-CMV-mCherry-2A-Cre, *n* = 12). *P*-values by two-tailed Student's *t* test. **d** Representative images of UCP1 and mCherry staining of scWAT in AAV-mCherry-2A-Cre-injected *Mypt1^{flox/flox}* mice (*n* = 3 per group). Black arrowheads, mCherry-positive and UCP1-negative; white arrowheads, mCherry- negative and UCP1-negative; yellow arrowheads, mCherry-positive and UCP-positive. Scale bar, 100 μm. **e** MYPT1 and ACTB immunoblotting of scWAT of AAV-injected *Mypt1^{flox/flox}* mice. **f** The body and scWAT

weights (*n* = 7 per group) of AAV-injected *Mypt1^{flox/flox}* mice. **g** *Mypt1^{+/flox}::Pdgfra*-Cre mice were exposed to 12 °C for 1 week (upper left). Immunoblotting of mitochondrial fraction of scWATs (lower left). Densitometric quantification of UCP1 immunoblot (right). *n* = 5 per group. P-JMJD1A immunoblotting of scWAT (inset). **h** Thermogenic gene mRNA levels in scWAT under RT (*Mypt1^{+/flox}*, *n* = 7; *Mypt1^{+/flox}::Pdgfra*-Cre, *n* = 6). **i** UCP1 staining of scWAT in *Mypt1^{+/flox}::Pdgfra*-Cre mice. **j** Changes of body weight (left) and body weight gain (right) in HFD-fed *Mypt1^{+/flox}* (*n* = 16) and *Mypt1^{+/flox}::Pdgfra*-Cre (*n* = 18) mice under RT. **k** Glucose tolerance test in HFD-fed mice in (**j**) (*Mypt1^{+/flox}*, *n* = 16; *Mypt1^{+/flox}::Pdgfra*-Cre, *n* = 17). **l** Fasting insulin levels in HFD-fed mice in (**j**) (*Mypt1^{+/flox}*, *n* = 14; *Mypt1^{+/flox}::Pdgfra*-Cre, *n* = 14). Line 2 of *Mypt1* floxed mice are used in (**a–l**) (see Methods and Supplementary Table 5). Data are mean ± SEM in (**b**, **f–h**, **j–l**). **b**, **f**, **g**, **h**, **j**, **k**, **l** Two-tailed Student's *t* test (log10-transformed data are used in (**c**) (left) and (**h**) to conform to normality). Source data are provided as a Source data file.

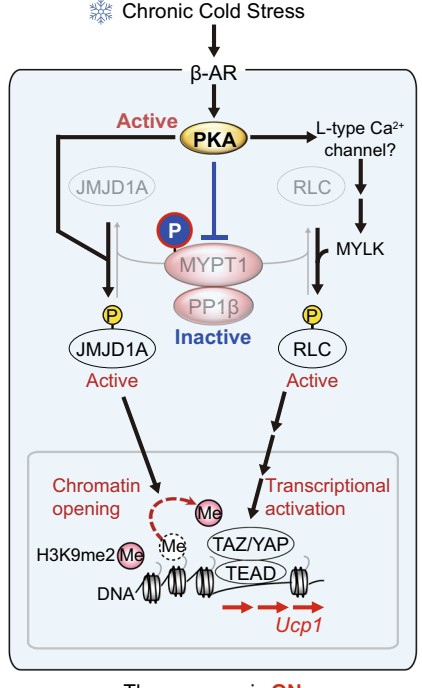

**Fig. 6 | MYPT1-PP1β-mediated mechanism for the regulation of thermogenic gene expressions in scWAT in cold or cold stress-free environment.** In cold environment, while PKA is activated, MYPT1-PP1β is inactivated through PKA-mediated MYPT1 phosphorylation, which in turn efficiently phosphorylates JMJD1A

and RLC and induces *Ucp1* expression through H3K9me2 demethylation and actomyosin-YAP/TAZ-mediated transcription. In cold stress-free environment, PKA is de-activated and inactive PKA leaves MYPT1-PP1β active (unphosphorylated form), which turns 'off' the phosphorylation and *Ucp1* transcription.

10-cm dishes, harvested, and centrifuged at 1000 × *g* for 5 min at 4 °C. Thereafter, the pellets of the cells were washed once with PBS, frozen in liquid nitrogen, and stored at −80 °C until use. Each cell pellet was sonicated on ice three times using 10 sec pulses with a Sonifier cell disruptor model 150D (Branson Ultrasonics), and thereafter the debris was removed by centrifugation at 20,000 × *g* for 10 min at 4 °C. Supernatants were filtered through a 0.2-μm filter and incubated with ANTI-FLAG M2 affinity gel (Sigma-Aldrich) for 2 h at 4 °C. The affinity gel was eluted three times with a 3X FLAG tag peptide (Sigma-Aldrich) for 10 min of wheel rotation at 4 °C. Next, each eluate was incubated with anti-human JMJD1A (F3640) cross-linked with Dynabeads-Protein G (Invitrogen), and rotated for 2 h at 4 °C. JMJD1A and/or its associated proteins were eluted using 0.1% RapiGest SF (Waters) in 50 mM NH$_4$HCO$_3$ buffer for 30 min at 60 °C. The eluent was concentrated using 10% ice-cold CCl$_3$COOH, washed with ice-cold acetone, dried, and resuspended in 25 vol% CH$_3$CN/25 mM NH$_4$HCO$_3$ at RT. The samples were reduced in 1.2 mM Tris (2-carboxyethyl) phosphine for 15 min at 50 °C and alkylated in 3 mM iodoacetamide for 30 min at RT. The samples were digested overnight with 100 ng trypsin (Promega) at

37 °C. After drying with a SpeedVac (Thermo Fisher Scientific), the peptides were dissolved in 0.2% CF$_3$COOH/2% CH$_3$CN and incubated for 60 min at 37 °C for residual degradation using RapiGest. After incubation, the samples were centrifuged to remove precipitates. Tandem mass spectrometry (MS) analysis was performed using an LTQ Orbitrap ELITE ETD mass spectrometer (Thermo Fisher Scientific). Aliquots of the trypsinized samples were automatically injected into a micro-pre-column, C18 PepMap 100 Peptide Trap cartridge (5 × 0.3 mm i.d.:: Thermo Fisher Scientific) attached to an injector valve for desalting and concentrating the peptides. After washing the trap with 98% H$_2$O/2% CH$_3$CN/0.2% CF$_3$COOH, the peptides were loaded onto a separation capillary reverse phase column (L-column2 micro C18 column 3 μm, 200 Å, 150 × 0.2 mm i.d.:: CERI, Tokyo, Japan) by switching the valve. The eluents used were A, 98% H$_2$O/2% CH$_3$CN/0.1% HCOOH, and B, 10% H$_2$O/90% CH$_3$CN/0.1% HCOOH. The column was developed at a flow rate of 1.0 μL/min, with a concentration gradient of CH$_3$CN: from 5% B to 35% B for 100 min, from 35% B to 95% B for 1 min, then sustained at 95% B for 9 min, from 95% B to 5% B for 1 min, and finally re-equilibrated with 5% B for 9 min. Effluents were introduced into the mass spectrometer

via a nanoelectrospray ion interface, which held the separation column outlet directly connected to a nanoelectrospray ionization needle (PicoTip FS360-50-30, New Objective Inc.). The electrospray ionization voltage was 2.0 kV, and the transfer capillary of the LTQ inlet was heated to 200 °C. No sheath or auxiliary gas was used. The mass spectrometer was operated in a data-dependent acquisition mode, in which the MS acquisition with a mass range of $m/z$ 420–1600 was automatically switched to MS/MS acquisition under the automated control of Xcalibur software. The top four precursor ions were selected by an MS scan with Orbitrap at a resolution of 240,000 for subsequent MS/MS scans by ion trap in the normal/centroid mode using the automated gain control mode with automated gain control values of $1 \times 10^6$ and $1.00 \times 10^4$ for full MS and MS/MS, respectively. We also employed a dynamic exclusion capability that allows the sequential acquisition of the MS/MS of abundant ions in the order of their intensities with an exclusion duration of 2.0 min and exclusion mass widths of −5 and +5 PPM. The trapping time was 100 ms with auto gain control on. Tandem mass spectra were extracted using the Proteome Discoverer version 2.1. All MS/MS samples were analyzed using Mascot (Matrix Science, version 2.6). Mascot was set up to search the SwissProt_2021_04 database (selected for *Mus*, 17,149 entries) with the digestion enzyme trypsin and a maximum number of missed cleavage sites of three. Mascot was searched with a product ion mass tolerance of 0.60 Da and a precursor ion tolerance of 5.0 ppm. The carbamidomethyl of cysteine was specified in Mascot as a fixed modification. Acetylation of the protein N-terminus, oxidation of methionine, pyroglutamation of glutamic acid, and phosphorylation of tyrosine, serine, and threonine are specified in Mascot as variable modifications. Scaffold (version Scaffol_5.1,, Proteome Software Inc.) was used to validate MS/MS-based peptide and protein identification. Peptide identifications were accepted if they could be established at greater than 95.0% probability. Protein identifications were accepted if they could be established at greater than 99.0% probability and contained at least three identified peptides.

### ChIP and ChIP-seq

The procedures for H3K9me2 assays using cultured cells have been described previously[1,9]. Briefly, cells were cross-linked with 0.5% formaldehyde for 10 min. Cross-linked cells were homogenized by passing them through a 22 G needle 10 times in ice-cold hypotonic buffer (10 mM HEPES-KOH, pH 7.5, 1.5 mM MgCl$_2$, 10 mM KCl, 1 mM EDTA, 1 mM EGTA containing 1 mM phenylmethylsulfonyl fluoride (PMSF), and protease inhibitor cocktail (Nacalai Tesque)) and centrifuged at $1000 \times g$ for 7 min to obtain a nuclear fraction. Cross-linked nuclear fractions were lysed in cell lysis buffer (23 mM Tris-HCl, pH 8.0, 3.0 mM EDTA, 0.9% Triton X-100, 134 mM NaCl, 1 mM PMSF, and protease inhibitor cocktail containing 0.2% SDS, sheared from 200 to 300 bp by using SONIFIER 250 (Branson) (output 4, duty cycle 60%, 20 s × 15 times) and used for immunoprecipitation. The amount of DNA in the nuclear lysate was determined by proteinase K (Sigma-Aldrich) protein digestion and DNA purification. 3 μg nuclear DNA containing nuclear lysates were immunoprecipitated for 2 h at 4 °C with 2 μg of anti-H3K9me2 antibody (IgG-6D11)[37] pre-bound to 50 μl of Dynabeads Protein G (Life Technologies). The immunoprecipitated DNA was subjected to qPCR with SYBR green fluorescent dye[1,9].

JMJD1A ChIP in cultured cells was performed as follows: Cells were cross-linked with 1.5 mM ethylene glycol bis (succinimidylsuccinate) (Thermo Scientific) for 30 min at RT, and then directly, a second crosslinking was performed through the addition of 1% formaldehyde for 10 min. After crosslinking, nuclear pellets were prepared for ChIP as described previously with the following modifications as described before[38,39]. Nuclear pellets were resuspended in 2 mL cell lysis buffer C containing 0.2% SDS, and chromatin DNA was sheared to 2 kb average in size through sonication. For ChIP, 30 μg nuclear DNA containing nuclear lysates were immunoprecipitated overnight at 4 °C with a mixture of two anti-JMJD1A antibodies (20 μg each of IgG-F0618 and IgG-F0231) pre-bound to 200 μL of Dynabeads protein G. Precipitated DNA from original DNA size of ~2 kb was further sheared to ~200 bp using an Acoustic Solubilizer (Covaris). Chromatin immunoprecipitated samples were used for library preparation using the KAPA Hyper Prep Kit (Nippon Genetics) according to the manufacturer's instructions[9,36]. Deep sequencing was performed on Illumina Genome Analyzer IIx using Sequencing Control Software v2.10.17 or HiSeq 2500 using HiSeq Control Software v2.2.58 as previously described[9,36]. Antibodies used: anti-H3K9me2 mouse mAb IgG-6D11 (Institute of Innovative Research, Tokyo Institute of Technology, Japan, 2 μg/mL for ChIP), anti-mouse JMJD1A mouse mAb IgG-F0618 (RCAST, The University of Tokyo, Japan, 25 μg/mL for ChIP), anti-mouse JMJD1A mouse mAb IgG-F0231 (RCAST, The University of Tokyo, Japan, 25 μg/mL for ChIP).

### ChIP-seq data processing

Trimmomatic[40] was used to filter low-quality reads and adapters from ChIP-seq reads in fastq files. ChIP-seq reads that passed quality filtering were aligned to the mouse mm9 genome using bowtie2[41]. Sam files generated by bowtie2 were then converted to sorted bam files using SAMtools[42]. PCR duplicates in the ChIP-seq reads were removed using Picard ((http://broadinstitute.github.io/picard/), a tool developed by the Broad Institute. Bigwig files of ChIP-seq were generated using the bamCoverage function of deeptools[43] whereas signals in the blacklist of mm9 were removed[44,45]. JMJD1A genomic binding regions (peaks) were calculated using the FindPeaks function of homer2[46]. Each JMJD1A genomic binding region was annotated to the nearest TSS using the annotatePeaks function of homer2. Motif searching of the JMJD1A genomic binding regions was performed using the findMotifsGenome.pl function of homer2.

### Quantitative real-time PCR (RT-qPCR) reaction

RT-qPCR was performed as described previously[47,48]. RT-qPCR was performed in 384-well plates using an ABI PRISM 7900HT sequence detection system (Applied Biosystems). All reactions were performed in triplicate. All the primer sequences used in this study are listed in Supplementary Tables 3 and 4.

### RNA sequencing (RNA-seq)

Total RNA was isolated using ISOGEN (Nippon Gene), according to the manufacturer's protocol. RNA-seq libraries were prepared using the TruSeq Sample Purification Kit (Illumina). Libraries were sequenced on a HiSeq 2500 platform (Illumina).

### Data processing of RNA-seq

Fastp[49] was used to filter out low-quality reads and adapters from the fastq files of the RNA-seq reads. After quality filtering, the RNA-seq reads were aligned to the mouse mm9 genome using STAR2[50]. RPKM tables of RNA-seq were calculated using GFOLD with GENOCODE comprehensive gene annotation set M1 release[51]. Pathway enrichment analysis of differentially expressed genes was performed using EnrichR[52] and homer2[46].

### Measurement of oxygen consumption

Oxygen consumption was measured using a Seahorse XF24 extracellular flux analyzer as described previously[9]. Briefly, the cultured adipocytes on day 7 of differentiation were detached with trypsin (0.5 g/L)/EDTA (0.53 mM) solution (Nacalai Tesque) and re-seeded at $1.0 \times 10^5$ cells per well into XF24 V7 cell culture microplates (Seahorse Bioscience). First, basal respiration was assessed in untreated cells. Second, NE-induced respiration was evaluated in response to 1 μM NE. To assess the oxygen consumption of mouse scWAT, scWAT was isolated, minced, and 6 mg of adipose tissue was placed into XF24 Islet

Capture Microplates (Seahorse Bioscience). The NE-induced oxygen consumption rate (OCR) was monitored after pre-incubation for 30 min in the pre-warmed XF24 assay medium supplemented with 2.5 μM NE.

## Phosphoproteomic analysis (isobaric tag labeling)

The cells were washed twice with PBS containing phosphatase inhibitors (40 mM NaF and 1 mM $Na_3VO_4$), harvested, and centrifuged at 400 × $g$ for 3 min at 4 °C. The obtained cell pellet was washed twice with TBS containing phosphatase inhibitors (40 mM NaF and 1 mM $Na_3VO_4$) (400 × $g$, 3 min, 4 °C), flash-frozen in liquid nitrogen, and stored at −80 °C until use. Frozen cell pellets were thawed out, resuspended in 0.2 mL of 120 mM sodium deoxycholate/120 mM sodium lauroylsarcosinate/1 M Tris-HCl, pH 9.0 containing 1% protease inhibitor and phosphatase inhibitor (Sigma-Aldrich), incubated on a heating block at 95 °C for 5 min, and then sonicated for 20 min in an ice-cold water bath. Proteins were quantified using a bicinchoninic acid protein assay kit, reduced with 10 mM dithiothreitol for 30 min at 37 °C, alkylated with 50 mM iodoacetamide in the dark for 30 min at 37 °C, digested with Lys-C (1/100 w/w, enzyme to protein, Wako) for 3 h at 37 °C, diluted 5-fold with 50 mM ammonium bicarbonate, followed by digestion with trypsin (1/100 w/w, enzyme to protein, Promega) overnight at 37 °C. An equal volume of ethyl acetate was added to the eluate and the mixture was acidified with 0.5% $CH_3COOH$. The mixture was shaken for 2 min, centrifuged at 15,800 × $g$ for 2 min, and then the aqueous phase was collected and desalted using StageTips with an SDB-XC Empore disk membrane (3 M), as described previously[53]. In brief, StageTips were prepared with 1 mL pipette tips packed with 3 disks (2.7 mm diameter), conditioned with 1 mL of 0.1% $CF_3COOH$/80% $CH_3CN$, and equilibrated using 1 mL of 0.1% $CF_3COOH$/5% $CH_3CN$. Approximately 100 mg of protein digest was loaded onto each Stage-Tip. The tips were then washed with 1 mL of 0.1% $CF_3COOH$/5% $CH_3CN$. Peptides were eluted from StageTips using 100 μL of 0.1% $CF_3COOH$/80% $CH_3CN$. Phosphopeptide enrichment by hydroxy acid-modified metal-oxide chromatography (HAMMOC)[54] using titania was performed as follows. A $C_8$ Empore disk with 0.6 mm diameter was inserted into a 0.1–10 μL pipette tip (Eppendorf). A slurry of bulk titania beads (0.5 mg) (titanium dioxide: particle size, 10 μm, GL sciences) in 20 μL of $CH_3OH$ was packed into the tip by centrifugation at 500 × $g$ for 5 min. Before loading samples, HAMMOC tips were equilibrated with 20 μL of buffer A (0.1% $CF_3COOH$/80% $CH_3CN$ with 300 mg/mL lactic acid (Wako) as a selectivity enhancer) by centrifugation at 1500 × $g$ for 5 min. 100 μg of the desalted tryptic digest in 0.1% $CF_3COOH$/80% $CH_3CN$ was diluted with 100 μL of buffer A and a 50 μL aliquot was loaded onto the HAMMOC tip four times by centrifugation at 1000 × $g$ for 5 min. After successive washing with buffer A and 0.1% $CF_3COOH$/80% $CH_3CN$ at 1500 × $g$ for 5 min, the phosphopeptide was eluted with 50 μL of 0.5% piperidine by centrifugation at 1000 × $g$ for 5 min. SDB-XC StageTips packed with RP-$C_{18}$ beads (0.25 mg) were activated with 0.1% $CF_3COOH$/80% $CH_3CN$ and equilibrated with 0.1% $CF_3COOH$/5% $CH_3CN$. The samples were loaded onto each tip and washed with 0.1% $CF_3COOH$/5% $CH_3CN$. Tandem mass tag (TMT) label reagents (Thermo Fisher Scientific) were dissolved in 5 μL of $CH_3CN$ and diluted with 95 μL of 50 mM phosphate buffer (pH 6.5). TMT solution (10 μg/10 μL) was loaded onto each tip. The samples were left on the reactor tip and labeled for 1 h at RT. After 1 h, the reactor tip was washed with 0.1% $CF_3COOH$/5% $CH_3CN$, and the peptides were eluted with 0.1% $CF_3COOH$/80% $CH_3CN$[55]. The sample was concentrated in a SpeedVac (Thermo Fisher Scientific), then resuspended in 10 μL of 0.5% $CF_3COOH$/4% $CH_3CN$ for subsequent nanoLC-MS/MS analysis. The data were collected using the SPS MS3 TMT method[56]. NanoLC-MS/MS analyses were performed using an Orbitrap system (Orbitrap Fusion Lumos, Thermo Fisher Scientific), a Dionex Ultimate 3000 RSLCnano system (Thermo Fisher Scientific), and an HTC-PAL autosampler (CTC Analytics). ReproSil C18 materials (3 μm, Dr. Maisch) were packed into an electrospray ionization needle (150 mm length × 100 μm I.D., 6 μm opening) to prepare an analytical column with "stone-arch" frit. The injection volume was 5 μL and the flow rate was 500 nL/min. The mobile phases consisted of (A) 0.5% $CH_3COOH$, and (B) 0.5% $CH_3COOH$ and 80% $CH_3CN$. A three-step linear gradient of 5% to 15% B for 5 min, 10% to 45% B for 100 min, 45% to 100% B for 5 min, and 100% B for 10 min was employed. A spray voltage of 2400 V was applied. The $MS^1$ scan range was 375–1500 $m/z$. The top ten precursor ions were selected in the $MS^1$ scan by the Orbitrap with resolution = 120,000 for $MS^2$ scans and the ion trap in automated gain control mode, where automated gain control values of $4.00 \times 10^5$ and $1.00 \times 10^4$ were set for full $MS^1$ and $MS^2$, respectively. To minimize repetitive $MS^2$ scanning, the dynamic exclusion time was set to 30 s, with a repeat count of 1. Normalized collision-induced dissociation was set to 35.0. $MS^3$ precursors were fragmented by high-energy collision-induced dissociation (CE = 65) and analyzed using Orbitrap (resolution of 50,000 and scan range of $m/z$ 100-500). A lock mass function was used for the Orbitrap Fusion Lumos to obtain constant mass accuracy during gradient analysis[57]. The database searches were performed as follows. Peak lists were created from the raw MS data using Proteo-Wizard v3.0.11018, ProteoDiscoverer v1.3.0.339, and MaxQuant v1.6.2.10 for Orbitrap Fusion Lumos based on the recorded fragmentation spectra. Peptides and proteins were identified by automated database searching using Mascot v2.4 (Matrix Science) against the SwissProt 2017_04 database (selected for *Mus*) with a precursor mass tolerance of 10 ppm, a fragment ion mass tolerance of 0.8 Da and strict trypsin specificity allowing for up to two missed cleavages. Carbamidomethylation of cysteine, TMT 6-plex of lysine, and TMT 6-plex of the protein N-terminus were set as fixed modifications, and the oxidation of methionine and phosphorylation of serine, threonine, and tyrosine were set as variable modifications. Peptides were considered identified if the Mascot score was over the 95% confidence limit based on the 'identity' score of each peptide and if at least three successive y- or b-ions with a further two or more y-, b- and/or precursor-origin neutral loss ions were observed, based on the error-tolerant peptide sequence tag concept. For unique phosphorylation site counting, phosphopeptides that overlapped on a protein were merged first to remove redundancy[58]. Phosphorylation site localization was evaluated using an in-house Perl script to check for the presence of a site-determining ion combination.

## Phosphoproteomic analysis (label-free quantification)

Immortalized pre-adipocytes established from scWAT seeded in 3.5-cm dishes were transfected with si-Ctrl or si-*Mypt1* #1 on the day of spreading for 48 h, washed twice with PBS containing phosphatase inhibitors (40 mM NaF and 1 mM $Na_3VO_4$), harvested, and centrifuged at 400 × $g$ for 3 min at 4 °C. Each obtained cell pellet was washed twice with TBS containing phosphatase inhibitors (40 mM NaF and 1 mM $Na_3VO_4$) and frozen at −80 °C until use. Each cell pellet was thawed out, lysed in SDS lysis buffer (5% SDS and 50 mM triethylammonium bicarbonate (TEAB), pH 7.55). Proteins were reduced by adding dithiothreitol to the protein solution in SDS to obtain a final concentration of 20 mM (10 min, 95 °C), alkylated by the addition of iodoacetamide to a final concentration of 40 mM (30 min, RT). The proteins were acidified to a final concentration of 1.2% phosphoric acid and diluted with six volumes of S-Trap buffer (90% $CH_3OH$ and 100 mM TEAB, pH 7.1) to aggregate proteins in colloidal particles. Tryptic digestion was performed using S-Trap Mini Spin columns (Protifi). Eluted peptides were dried in a SpeedVac (Thermo Fisher Scientific) and resuspended in 60 μL of 0.1% TFA/2% $CH_3CN$ solvent. Phosphopeptide enrichment was performed at RT using a micro-column tip packed with 1 mg of $TiO_2$ (Titansphere Phos-TiO MP Kit, GL Sciences). The resin was conditioned with 20 μL of buffer B (0.4 vol% $CF_3COOH$, 80 vol% $CH_3CN$) (3000 × $g$, 2 min) and equilibrated with 40 μL of buffer C (0.3 vol% $CF_3COOH$, 60 vol% $CH_3CN$, and 25 vol%

lactic acid) (3000 × *g*, 2 min). The eluted digests (50 μL) and an equal volume of buffer C were mixed and passed through the spin tip twice (1000 × *g*, 10 min). The spin tip was rinsed with 20 μL of buffer C and 20 μL of buffer B (3000 × *g*, 2 min for each 20 μL). Phosphopeptides, mostly containing mono-phosphorylated peptides, were eluted from the spin tip with 500 μL of buffer D (20 mM methylphosphoric acid and 20 vol% $CH_3CN$) (2 × 250 μL aliquots at 3000 × *g*, 10 min per aliquot). Thereafter, the phosphopeptides mostly containing multi-phosphorylated peptides were eluted from the spin tip (1000 × *g*, 5 min) using successive elution with (i) 100 μL of 500 mM $Na_2HPO_4$, (ii) 50 μL of 5 vol% ammonia solution, and (iii) 50 μL of 5 vol% pyrrolidine. Phosphopeptides eluted with buffer D were desalted using a Mono-Spin C18 column (GL Sciences). The resin was conditioned with 500 μL of buffer E (0.1% $CF_3COOH$, 80 vol% $CH_3CN$) (5000 × *g*, 2 min, RT) and equilibrated with 500 μL of buffer F (0.1% $CF_3COOH$, 5 vol% $CH_3CN$) (5000 × *g*, 2 min, RT). Phosphopeptides eluted with buffer D were diluted five times with 20 vol% $CF_3COOH$ and passed through a spin column (1000 × *g*, 2 min, RT) until all samples had passed through the spin column. The spin column was rinsed with 1000 μL of buffer E (1000 × *g*, 2 min, RT). The phosphopeptides were eluted from the spin column with 250 μL of buffer E (1000 × *g*, 2 min, RT). Eluates were dried in a SpeedVac (Thermo Fisher Scientific) and resuspended in buffer F. Phosphopeptides eluted with $Na_2HPO_4$, ammonia solution, or piperidine were desalted using GL-Tip SDB (GL Sciences). The resin was conditioned with 20 μL of buffer E (3000 × *g*, 2 min, RT) and equilibrated with 20 μL of buffer F (3000 × *g*, 2 min, RT). Phospho-peptides eluted with $Na_2HPO_4$, ammonia solution, or piperidine were diluted twice with 20 vol% $CH_3COOH$ and passed through a spin column (3000 × *g*, 5 min, RT) until all samples had passed through the spin column. The spin column was rinsed with 20 μL of buffer F (3000 × *g*, 2 min, RT). Phosphopeptides were eluted from the spin column with 50 μL of buffer E using a syringe. Eluates were dried in a SpeedVac (Thermo Fisher Scientific) and resuspended in 0.1% TFS/2% $CH_3CN$. A capillary reverse-phase high-performance liquid chromatography-MS/MS system (ZAPLOUS System: AMR Inc.), comprising of an advanced ultra-high-performance liquid chromatography instrument (Michrom Bioresources, Auburn, CA, USA), an HTC PAL autosampler (CTC Analytics), an Orbitrap Fusion ETD, and a quadrupole linear ion trap Orbitrap mass spectrometer (Thermo Fisher Scientific) equipped with a Dream Spray ESI source (Dream Spray, AMR), was used for LC-MS/MS analysis. Aliquots of samples were automatically injected onto a micro-precolumn C18 PepMap 100 Peptide Trap cartridge (5 × 0.3 mm i.d.:: Thermo Fisher Scientific) attached to an injector valve for desalting and concentrating the peptides. After washing the trap with 98% $H_2O$/2% $CH_3CN$/0.2% $CF_3COOH$, the peptides were loaded into a separation capillary reverse phase column (L-column2 micro C18 column 3 μm, 200 Å, 150 × 0.2 mm i.d.:: CERI, Tokyo, Japan) by switching the valve. The eluents used were: A, 99.9% $CH_3CN$/0.1% HCOOH:: and B, 100% $CH_3CN$. The column was developed at a flow rate of 1.0 μL/min, with a concentration gradient of $CH_3CN$: from 5% B to 30% B for 100 min, then from 30% B to 95% B for 1 min, maintained at 95% B for 9 min, from 95% B to 5% B for 1 min, and finally re-equilibrated with 5% B for 9 min. Effluents were introduced into the mass spectrometer via a nanoelectrospray ion interface that held the separation column outlet directly connected to a Dream Spray electrospray ion source. The electrospray ionization voltage was 1.5 kV and the transfer capillary of the orbitrap inlet was heated to 250 °C. No sheath or auxiliary gas was used. The mass spectrometer was operated in a data-dependent acquisition mode, in which the MS acquisition with a mass range of *m/z* 390-1,590 was automatically switched to MS/MS acquisition under the automated control of Xcalibur software. The top 20 precursor ions were selected by an MS scan with Orbitrap at a resolution of 240,000, and for the subsequent MS/MS scans through ion trap in the normal/centroid mode, using the AGC mode with AGC values of 4 × $10^5$ and 5 × $10^3$ for full MS and MS/MS, respectively. We also employed a dynamic

exclusion capability that allowed sequential acquisition of the MS/MS of abundant ions in the order of their intensities with an exclusion duration of 5 s and exclusion mass widths of −5 and +5 ppm. The trapping time was 35 ms with auto gain control on. Tandem mass spectra were extracted using the Proteome Discoverer version 2.1. All MS/MS samples were analyzed using Mascot (Matrix Science, version 2.6.1). Mascot was set up to search the SwissProt_2019_04 database (selected for *Mus*, 17,073 entries) with the digestion enzyme trypsin and a maximum of three missed cleavage sites. Mascot was searched with a product ion mass tolerance of 0.60 Da and a precursor ion tolerance of 5.0 ppm. The carbamido-methyl of cysteine was specified in Mascot as a fixed modification. Acetylation of the protein N-terminus, oxidation of methionine, pyroglutamation of glutamate, and phosphorylation of tyrosine, serine, and threonine were specified in Mascot as variable modifications. Scaffold (version Scaffold_4.8.9, Proteome Software Inc.) was used to validate MS/MS-based peptide and protein identification. Peptide identifications were accepted if they could be established at greater than 95.0% probability.

### Database search for identification of kinase consensus sequences

Putative PKA phosphorylation sequences located in MYPT1 were surveyed using Scansite Motif Scanner (https://scansite4.mit.edu/#home).

### Mice

We used two lines of *Mypt1* floxed mice as summarized in Supplementary Table 5. *Mypt1* floxed mice were generated and genotype analysis of the flox locus was performed as described previously (line 1)[26]. We generated another line of *Mypt1* floxed mice (line 2) as follows: A CRISPR-Cas9 expression vector, sgRNAs targeting *Mypt1* intron 5 (5′-GCAGCAGTGCTTACGTTAGGCGG-3′) and *Mypt1* intron 8 (5′-GCAATGATGAGGGTTTAAAGTGG-3′), and donor plasmid DNA including *Mypt1* Exon 6-8 flanked by two lox P sites were injected into fertilized C57BL/6J eggs. The detailed method for harvesting fertilized eggs has been described elsewhere[59]. PCR-based screening and Sanger sequencing were performed to identify the founder mice. For genotyping, clipped toes were used for PCR. Genotyping was performed using primers flanking the LoxP site located in *Mypt1* intron 8 (forward primer: 5′-TCCAAAGAGGCAATGATGAGGG-3′; reverse primer: 5′-TGC AACCATCCAGCTTGGTA-3′). PCR products of 144 and 206 bp were amplified from *WT* and floxed alleles, respectively. *Pdgfrα*-Cre (stock 013148) mice were purchased from Jackson Laboratories. *Adipoq*-Cre mice were generated as described previously[27]. Genotyping of *Adipoq*-Cre and *Pdgfrα*-Cre mice was performed using primers specific for the Cre transgene, forward primer: 5′-GAACCTGATGGACATGTTCAGG-3′ and reverse primer: 5′-AGTGCGTTCGAACGCTAGAGCCTGT-3′ to detect Cre expression (product size, 320 bp). To generate *Mypt1* + / *flox::Adipoq-Cre* mice, in which *Mypt1* is specifically ablated in adipocytes, we crossed *Mypt1* floxed mice (line 1) with *Adipoq*-Cre mice In addition, for scWAT culture experiments, *Mypt1* floxed mice (line 1) were used. To generate *Mypt1* + /*flox::Pdgfra-Cre mice*, in which *Mypt1 is specifically ablated i*n pre-adipocytes we crossed *Mypt1* floxed mice (line 2) with *Pdgfra*-Cre mice. For AAV injection-mediated depletion of MYPT1 in scWAT of mice, control AAV (AAV-CMV-mCherry) or AAV expressing Cre recombinase (AAV-CMV-mCherry-2A-Cre) were injected into the right or left pad of the scWAT of *Mypt1* floxed mice (line 2), respectively, following a published protocol[60]. In short, AAV was injected into the scWAT of *Mypt1*^*flox/flox*^ mice at a viral titer of 1.5 × $10^{10}$ genomic copies (GC) per pad. 30 μL of AAV at a dose of 0.5 × $10^{10}$ GC/μL was injected into each scWAT (5 μL per injection, 6 locations per pad). scWAT was harvested two weeks after viral infection, and the efficacy of knockdown was evaluated through the quantification of *Mypt1* expression levels.

## Animal experiments

All animal experiments followed the basic guidelines for conducting animal experiments established by the Animal Care and Use Committee of the University of Tokyo and Tohoku University. Mice were maintained in a temperature- and humidity-controlled environment under a 12 h light/12 h dark cycle (08:00-20:00) at constant temperature (23 °C) with free access to food and water. Animals were fed a normal chow diet (CE-2, CLEA Japan Inc.) from the age of 4 weeks or HFD, consisting of 58.0% fat, 15.0% protein, and 27.0% carbohydrate, from the age of 5 to 7 weeks as described previously[61-63]. For chronic cold exposure experiments, animals were single-caged and exposed to 12 °C for one week. At the conclusion of experiments, tissues were harvested and snap frozen in liquid nitrogen for RNA and protein analysis. For cold tolerance test, mice housed at RT for 1 week prior to the experiments were exposed to 8 °C from 10 a.m., and core body temperature was monitored using a rectal thermometer (Bio Research Center: BDT-100C). Details of the age and sex of mice used are shown in Supplementary Table 5.

## Blood parameters

Blood glucose concentrations were measured using LAB Gluco (ForaCare). Plasma insulin levels were determined by enzyme-linked immunosorbent assay using the Insulin Immunoassay Kit (Shibayagi) according to the manufacturer's instructions.

## GTT and GSIS

For GTT and GSIS, mice were fasted for 6 h and then administered an intraperitoneal injection of glucose (1.5 g/kg BW)[10]. Blood samples for glucose and insulin measurements were obtained from the tail vein at the indicated time points.

## Isolation of nuclear and mitochondrial fraction from scWAT

Isolation of nuclear and mitochondrial fractions from scWAT was performed using the Mitochondrial Isolation Kit for Tissue (Abcam) according to the manufacturer's instructions. Frozen tissues were washed with wash buffer, homogenized with a Potter-Elvehjem homogenizer in ice-cold isolation buffer containing both protease inhibitors (2.8 µg/mL aprotinin, 5 µg/mL pepstatin A, 10 µg/mL leupeptin, 1 mM PMSF, 1 mM DTT, 1 mM benzamidine) and phosphatase inhibitors (40 mM NaF, 1 mM $Na_3VO_4$, 10 mM β-glycerophosphate), and centrifuged at 1000 × $g$, 4 °C, for 10 min, to obtain post-nuclear supernatant and nuclear pellet. The post-nuclear supernatant was centrifuged at 12,000 × $g$, 4 °C, for 15 min to obtain mitochondrial pellets. Pellets were washed once with isolation buffer and lysed in Buffer A (20 mM HEPES-NaOH (pH 7.5), 1 mM EDTA, 150 mM NaCl, 0.1% SDS, 0.5% sodium deoxycholate, 1% Triton X-100) containing both protease inhibitors (2.8 µg/mL aprotinin, 5 µg/mL pepstatin A, 10 µg/ mL leupeptin, 1 mM PMSF, 1 mM DTT, 1 mM benzamidine) and subjected to SDS-PAGE. Nuclear pellets were lysed in Buffer B (50 mM HEPES-KOH (pH 7.9), 500 mM NaCl, 1.5 mM $MgCl_2$, 0.1 % NP-40) containing both protease inhibitors (2.8 µg/mL aprotinin, 5 µg/mL pepstatin A, 10 µg/mL leupeptin, 0.5 mM PMSF) and phosphatase inhibitors (40 mM NaF, 1 mM $Na_3VO_4$, 10 mM β-glycerophosphate), centrifuged at 22,000 × $g$, 4 °C, for 20 min. The supernatant was further centrifuged at 120,000 × $g$, 4 °C, for 30 min, and the supernatant was collected as nuclear lysate. The nuclear lysate was subjected to immunoprecipitation using an anti-JMJD1A antibody in buffer B for 2 h at 4 °C to detect JMJD1A phosphorylation by immunoblotting.

## Immunohistochemistry

scWAT isolated from mice was placed in 4% paraformaldehyde for 48 h, stored in 70% ethanol, and subsequently embedded in paraffin. Immunostaining was performed on deparaffinized 3.5 µm sections, which were rehydrated; the endogenous peroxidase activity was quenched by treatment with 1% or 0.3% hydrogen peroxide for 20 min

for mCherry or UCP1 staining, respectively. Following antigen epitope retrieval, performed by autoclaving the slides in antigen retrieval solution (pH 9), the sections were incubated overnight at 4 °C with mouse monoclonal anti-mCherry antibody (ab125096, Abcam) at a dilution of 1:150, or with rabbit polyclonal anti-UCP1 antibody (ab23841, Abcam) at a dilution of 1:5000. The bound anti-mCherry or anti-UCP1 was detected using Simple Stain™ mouse or rabbit MAX PO (NICHIREI), respectively. Bound horseradish peroxidase-labeled polymers were detected by the addition of a 3,3-diaminobenzidine substrate-chromogen solution and counterstained with hematoxylin. Slides were scanned using VS200 slide scanner (OLYMPUS).

## Adenovirus infection

Primary pre-adipocytes isolated from the SVF of the scWAT of $Mypt1^{flox/flox}$ mice were seeded, cultured for 24 h, infected with either AxCAN-LacZ or AxCAN-LacZ at a multiplicity of infection of 900 for 24 h (day −1 to day 0), and then induced for differentiation as described above.

## Isolation of SVFs and mature adipocytes

Isolation of SVF and mature adipocytes was performed as described previously[64] with slight modifications. Briefly, scWAT of *WT* mice injected with AAV (AAV-CMV-mCherry) was excised, minced in Krebs-Ringer bicarbonate HEPES (KRBH) buffer (120 mM NaCl, 4 mM $KH_2PO_4$, 0.75 mM $CaCl_2$, 1 mM $MgSO_4$, 10 mM $NaHCO_3$, 30 mM HEPES, pH 7.4) containing 2% fatty acid-free bovine serum albumin (Sigma-Aldrich), 25 mM glucose, and 2 mg/mL collagenase (Wako), and incubated at 37 °C for 1 h with shaking at 150 cycles/min. After centrifugation at 1700 × $g$ for 10 min at 25 °C, the floating mature adipocytes were collected in ISOGEN (Nippon Gene). Centrifuged cell pellets, termed SVF, were resuspended in KRBH buffer, filtered through a 100 µm nylon cell strainer (BD Falcon), and centrifuged at RT at 1700 × $g$ for 10 min at 25 °C. Centrifuged cell pellets were collected using ISOGEN (Nippon Gene).

## Statistical analysis

Data are presented as the mean ± SEM for independent experiments. Comparisons between two groups were performed using the paired t-test or the Student's $t$ test, as appropriate. Two-way repeated measures ANOVA was used to compare repeated measurements. The Wilcoxon–Mann–Whitney test was used to compare the sum of peak scores or distances of peaks and nearest TSS between the two groups. $P$-values denoted as *$P < 0.05$; **$P < 0.01$, and ***$P$-value < 0.005.

## Reporting summary

Further information on research design is available in the Nature Research Reporting Summary linked to this article.

# Data availability

Data supporting this study are available from the corresponding authors upon reasonable request. RNA-seq transcriptome data and JMJD1A ChIP-seq data (day 0 and day 4) were deposited in the Gene Expression Omnibus (GEO) database with accession number GSE202506. JMJD1A ChIP-seq data (day 8) has already been published and deposited in the GEO database GSE107901. Proteomic data were deposited in ProteomeXchange Consortium under the accession number PXD031210, PXD031896, and PXD031897. The *Mus* SwissProt database (v. 2017_04, 2019_04, and 2021_04) was used for proteome analyses. Source data are provided with this paper.

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

## Acknowledgements

The authors thank Dr. Evan D Rosen for *Adipoq*-Cre mice; Dr. Wataru Ogawa and Dr. Tao Tao for helpful discussions regarding mice experiments; Dr. Toshio Kitamura for pMXs-puro plasmid; Dr. Akihiro Yamanaka for pAAV plasmid; Dr. Shogo Yamamoto for analyzing RNA-seq data; Akashi Taguchi-Izumi for ChIP/RNA-seq assistance; Aya Nakayama for phosphoproteomics assistance; Minori Yoshio, and Yasuyo Ono for technical and secretary assistance; and Dr. Kazuhisa Takeda, Toku Nishiyama, Wakako Endo, Shun Kakinuma, Daichi Akiba and all the members of the Sakai laboratory for helpful discussions. The supercomputing resource was provided by Human Genome Center (the Univ. of Tokyo) and by Tohoku Medical Megabank Organization (Tohoku Univ.). We also thank Dr. Satoru Takahashi and Dr. Seiya Mizuno of Laboratory Animal Resource Center in Transborder Medical Research Center, the University of Tsukuba for creating *Mypt1* floxed mice. We are grateful to the Biomedical Research Core of Tohoku University Graduate School of Medicine for supporting the immunohistochemical experiments. This study was supported by grants in-aid for scientific research (JP16H06390, JP21H04826), for scientific research on innovative areas (JP20H04835), for Exploratory Research (JP20K21747) (to J.S.), for research activity start-up (JP21K21211), for JSPS Fellows (JP19J11909) (to H.T.) from the Ministry of Education, Science, Sports and Culture (MEXT), by AMED-CREST under Grant Number JP20gm1310007 (to Y.M., T.Y., and J.S.), and by SECOM Science and Technology Foundation.

## Author contributions

T.Y., Y.A, R.I., and C.Y. contributed equally to this work. Y.M. and J.S. designed the study and wrote the paper. T.F.O. critically commented and edited the paper. H.T., G.Y., T.Y., R.I., Y.A., A.U., R.Y., Y.N., K.Y., M.O.-K, J.H., H.C., M.T., S.X., J.Z., H.S., and Y.Z. performed experiments. J.N., Y.I. and T.Kawamura. performed proteomics analysis. H.A. and Y.W. supported ChIP-seq, and C.Y. and Y.M. carried out analyses. H.K., M.-S.Z., and M.A. provided materials. Y.S., S.Y., T.T., T.I., and T.Kodama. commented on the paper. All authors reviewed the results and approved the final version of the manuscript.

## Competing interests

The authors declare no competing interests.

## Additional information

Hiroki Takahashi[1,2,12], Ge Yang[1,12], Takeshi Yoneshiro[2], Yohei Abe[2], Ryo Ito[1], Chaoran Yang[1], Junna Nakazono[3], Mayumi Okamoto-Katsuyama[2], Aoi Uchida[2], Makoto Arai[1], Hitomi Jin[1], Hyunmi Choi[1], Myagmar Tumenjargal[1], Shiyu Xie[1], Ji Zhang[1], Hina Sagae[1], Yanan Zhao[1], Rei Yamaguchi[1], Yu Nomura[1], Yuichi Shimizu[1], Kaito Yamada[1,4], Satoshi Yasuda[4], Hiroshi Kimura[5], Toshiya Tanaka[6], Youichiro Wada[7], Tatsuhiko Kodama[6], Hiroyuki Aburatani[8], Min-Sheng Zhu[9], Takeshi Inagaki[2,10], Timothy F. Osborne[11], Takeshi Kawamura[7], Yasushi Ishihama[3], Yoshihiro Matsumura[1,2]✉ & Juro Sakai[1,2]✉

[1]Division of Molecular Physiology and Metabolism, Tohoku University Graduate School of Medicine, Sendai 980-8575, Japan. [2]Division of Metabolic Medicine, Research Center for Advanced Science and Technology, The University of Tokyo, Tokyo 153-8904, Japan. [3]Department of Molecular and Cellular BioAnalysis, Graduate School of Pharmaceutical Sciences, Kyoto University, Kyoto 606-8501, Japan. [4]Department of Cardiovascular Medicine, Tohoku University Graduate School of Medicine, Sendai 980-8574, Japan. [5]Cell Biology Center, Institute of Innovative Research, Tokyo Institute of Technology, Yokohama 226-8503, Japan. [6]Department of Nuclear Receptor Medicine, Laboratories for Systems Biology and Medicine, Research Center for Advanced Science and Technology, The University of Tokyo, Tokyo 153-8904, Japan. [7]Isotope Science Center, The University of Tokyo, Tokyo 113-0032, Japan. [8]Genome Science and Medicine Division, Research Center for Advanced Science and Technology, The University of Tokyo, Tokyo 153-8904, Japan. [9]Model Animal Research Center and Ministry of Education (MOE) Key Laboratory of Model Animal for Disease Study, Nanjing University, 210061 Nanjing, China. [10]Laboratory of Epigenetics and Metabolism, Institute for Molecular and Cellular Regulation, Gunma University, Gunma 371-8512, Japan. [11]Institute for Fundamental Biomedical Research, Johns Hopkins All Children's Hospital, and Medicine in the Division of Endocrinology, Diabetes and Metabolism of the Johns Hopkins University School of Medicine, 600 Fifth Street S. St., Petersburg, FL 33701, USA. [12]These authors contributed equally: Hiroki Takahashi, Ge Yang. ✉e-mail: matsumura-y@lsbm.org; jmsakai@med.tohoku.ac.jp

