## [Peer Review File · Nature Communications]

MYPT1-PP1 β phosphatase negatively regulates both chromatin landscape and co-activator recruitment for beige adipogenesisREVIEWER COMMENTS

Reviewer #1 (Remarks to the Author):

In the present manuscript, Takahashi et al., investigated the epigenetic regulation during beige adipogenesis using several unbiased approaches, including proteomics, ChIP-seq and transcriptomics, and in vivo mice models. They demonstrated that MYPT1-PP1beta phosphatases inhibited JMJD1A and myosin RLC, leading to suppress beige adipogenesis. They further showed that deficiency of Mypt1 increased activities of JMJD1A and RLC, which in turn, remove suppressive histone markers and activated TAZ/YAP downstream signaling. Although the in vitro data on the relationship between MYPT1-PP1beta and JMJD1A appeared to be convincing, more data from in vivo analyzes should be provided to support the idea. Further, the authors should improve the description of the results and methods. Detailed comments are following.

Major comments

1. In the in vivo experiments (Figs. 5a, 5d, Supplementary Fig. 5f), the effects of Mypt1 deficiency on beige adipogenesis should be further carefully examined by several experiments, including UCP1 immunoblotting, lipid droplet morphology, and gene expression profiles. In addition, the authors should test whether mCherry-positive cells would reflect an increased number of beige adipocytes compared to mCherry-negative cells using AAV-mCherry-2A-Cre-injected mice.
2. Although the authors showed that Ucp1 expression was regulated by the MYPT1-JMJD1A axis, the effect of MYPT1 deficiency on H3K4me2 is still unclear. They should investigate the levels of H3K4me2 in the Ucp1 enhancer upon overexpression of demethylation defective JMJD1A mutant or WT-JMJD1A (related to Figs. 3h and 3i). Moreover, the levels of H3K4me2 in Mypt1-deficient mice were not determined. Also, they should examine whether the histone methylation of Ucp1 enhancer is modulated by JMJD1A in these experimental conditions.
3. In Fig. 5, what are the levels of JMJD1A and RLC phosphorylation in Mypt1 +/-flox::Pdgfra-Cre mice? They should verify whether MYPT1-PP1beta dephosphorylates JMJD1A and RLC in vivo in the context of beige adipogenesis during cold exposure.
4. In this study, the authors assumed that Ucp1 mRNA expression (Figs. 2e, 2h, 2l, 3d, 3h, 3l, 4h) would be the marker of beige adipogenesis. Not only Ucp1 but also other thermogenic genes and beige marker genes should be investigated.
5. The authors claimed that PKA inhibited MYPT1/PP1beta and, in turn, activated JMJD1A, eventually promoting beige adipogenesis upon cold exposure. Recent studies have shown that turn-off of PKA signaling is involved in beige-to-white transition upon re-warming stimuli (PMID: 29657031, 29692364, 27568548). In this regard, the importance of this study would be further strengthened when they examine whether MYPT1 is involved in beige-to-white transition.
6. The authors suggested that MYPT1 would play suppressive roles in preadipocytes, not in mature beige adipocytes, which was supported by Adipoq-Cre Mypt1 KO mice model (Supplementary Figs. 5f-i). However, given that PKA signaling, JMJD1A, and MYPT1/PP1beta are also present in mature adipocytes, appropriate explanations of the different roles of MYPT1/PP1beta in preadipocytes and adipocytes should be provided.

Minor comments

1. Many graphs do not display n numbers and statistics.
2. In Figs. 1m and 3e, how the relationship between the binding site of JMJD1A and gene expression was investigated? What is the rationale for setting the standard to 50 kb?
3. For the MS/MS data reports (Figs. 1b, 1c), more details such as peptide numbers and protein match

percentage should be included.

4. The interpretation of the oil-red O experiment is puzzling. It was not fully explained in the text.
5. In Figs. 5e, 5f, 5g, the number of biological replicates appears to be small.
6. Provide biological replicates and mice information for all experiments.
7. The western blot in Figs. 1d and 4g were performed individually, suggesting either inconsistent or incomparable. These blots should be reperformed in the same blot.
8. In place of bar graphs, small datasets show the full data as univariate scatter plots that also show mean and standard deviation (Fig. 1i, 1j, 1k, 2e, 2i, 3d, 3h, 3i).

Reviewer #2 (Remarks to the Author):

The manuscript titled "MYPT1-PP1B phosphatase negatively regulates both chromatin landscape and co-activator recruitment to influence gene expression during beige adipogenesis" by Takahashi et al reported a previously unknown regulator MYPT1-PP1B in beige fat biology. The group has previously reported that a histone H3 lysine 9 (H3K9) demethylase JMJD1A plays an important role regulating beige fat function, and is activated through phosphorylation at serine 265 by protein kinase A (Nature Communications 2015, Abe et al. and Nature Communications 2018 Abe et al). Current study is a nice follow-up to reveal the identity of the phosphatase that is responsible for the dephosphorylation of s265 in JMJD1A in beige fat. The unbiased experimental approach was well controlled and presented data were convincing. In addition, the author further uncovered an interesting connection between MYPT1-PP1B to myosin regulatory light chain (RLC) regulated pathway, which mediates beige fat function through TAZ/YAP. Even though this is continuation of previous study, the novelty and significance of the present study is evident.

The manuscript is well written and most of the results are of high quality, strongly supporting proposed hypothesis. In particular, the authors demonstrated that general adipogenesis is not affected by MYPT1-PP1B, rather its effects are specifically affecting thermogenic function. This separates current study from many other reports that claim protein X or Y are thermogenic fat regulators but most likely they just affect fat function in general.

This reviewer does have some comments/suggestions for data shown in figure 5, in particular the data using Mypt1+/flox;pdgfra-cre mice.

As of right now, there is no perfect Cre line for general preadipocytes, nor for beige fat precursor. PDGFRA-CRE has been used as preadipocyte cre line in many studies, but its caveats were well documented. Indeed, the fact that Mypt1flox/flox;pdgfra-cre mice die embryonically suggests that MYPT1 in other cell types in addition to preadipocyte may play an essential role in development. The drastic phenotype shown in fig5E and fig5g from heterogeneous Mypt1 in PDGFRA-lineage is a bit complicated to interpret. The most responsive organs upon acute cold exposure are skeletal muscle which regulates shivering and classical brown fat which will be immediately activated upon cold. In comparison, beige adipogenesis within subcutaneous fat upon cold exposure, the full extent of activation may take a long time and involve tissue remodeling, including new precursor for beige adipocytes to differentiate. The fact that after only 8 hrs, there is body weight change and subcutaneous adipose tissue mass change and body temperature change, it is unlikely caused by half dose of mypt1 in just beige precursors. Further exploration should be carried out. Alternatively, the authors may consider to just remove this model from the current study, since there are several other in vivo models, which are better controlled with fewer caveats and also support the hypothesis.

Minor point:

1. In data availability section, the authors indicate that the RNA-seq transcriptome data and JMJD1A

ChIP-seq data have been deposited at GEO but the accession number was listed as "GSEXXXXX". Customarily, most authors deposit these data as private data protected with a password, with public release set upon publication date of the manuscript. An actual GSE number should be included, and a password should be made available to reviewers upon request.

Responses to reviewer's comment

Reviewer #1: *In the present manuscript, Takahashi et al., investigated the epigenetic regulation during beige adipogenesis using several unbiased approaches, including proteomics, ChIP-seq and transcriptomics, and in vivo mice models. They demonstrated that MYPT1-PP1beta phosphatases inhibited JMJD1A and myosin RLC, leading to suppress beige adipogenesis. They further showed that deficiency of Mypt1 increased activities of JMJD1A and RLC, which in turn, remove suppressive histone markers and activated TAZ/YAP downstream signaling. Although the in vitro data on the relationship between MYPT-PP1beta and JMJD1A appeared to be convincing, more data from in vivo analyzes should be provided to support the idea. Further, the authors should improve the description of the results and methods. Detailed comments are following.*

Major comments

Major comment 1: *In the in vivo experiments (Figs. 5a, 5d, Supplementary Fig. 5f), the effects of Mypt1 deficiency on beige adipogenesis should be further carefully examined by several experiments, including UCP1 immunoblotting, lipid droplet morphology, and gene expression profiles (**major comment 1-1**). In addition, the authors should test whether mCherry-positive cells would reflect an increased number of beige adipocytes compared to mCherry-negative cells using AAV-mCherry-2A-Cre-injected mice (**major comment 1-2**).*

Reply to major comment 1-1: We appreciate the valuable comment made by the reviewer. Regarding **Fig. 5a** (*in vivo* experiments using *Mypt1^{lox/lox}* mice injected with AAV-mCherry-2A-Cre), we additionally, performed i) UCP1 immunoblotting, ii) histological analysis (lipid droplet morphology), and (iii) gene expression profiles of thermogenic genes from the scWAT of *Mypt1^{lox/lox}* mice injected with either control AAV-mCherry or AAV-mCherry-2A-Cre acclimated to 12 °C.

(i) Immunoblotting showed that UCP1 protein levels were significantly elevated in scWATs of mice injected with AAV-mCherry-2A-Cre compared to AAV-mCherry (**Fig. 5b**). AAV-mCherry-2A-Cre injection slightly decreased scWAT weight without affecting body weight (**Fig. 5f**).

Fig. 5b UCPI immunoblotting of the mitochondrial fraction of scWATs from *Mypt1^{lox/lox}* mice (line 2) injected with control AAV-mCherry or AAV-mCherry-2A-Cre (left). Densitometric quantification of UCPI immunoblotting (right). TOM20 was used as a mitochondrial loading control. n = 7 per group.

Fig. 5f The body and scWAT weights (n = 7 per group) of scWAT of AAV injected *Mypt1^{lox/lox}* mice (line 2) after 1 week acclimation to 12°C

(ii) Gene expression profiles: qPCR analysis showed that the expression of not only *Ucp1* but also other thermogenic genes (e.g., *Prdm16*, *Cidea*, *Otop1*) was significantly elevated in scWATs injected with AAV-mCherry-2A-Cre compared to those injected with AAV-mCherry (**Fig. 5c**).

Fig. 5c Expression of thermogenic genes in the scWAT of AAV-injected *Mypt1^{lox/lox}* mice (line 2) after 1 week of acclimation to 12 °C. Data are presented as mean ± SEM (n = 10 per group, left). *P*-values by paired t-test. NE-induced OCR normalized to tissue mass in scWAT AAV-injected *Mypt1^{lox/lox}* mice after 1 week of acclimation to 12 °C (right). Data are presented as the mean ± SEM (AAV-CMV-mCherry, n = 10; AAV-CMV-mCherry-2A-Cre, n = 12). *P*-values were calculated using Student's t-test.

(iii) Lipid droplet morphology: Hematoxylin & eosin staining shows that scWATs of mice injected with AAV-mCherry-2A-Cre and housed 12 °C for 1 week contained multilocular lipid droplets containing adipocytes expressing UCP1 (**Supplementary Fig. 5e**). In contrast, injection of AAV-mCherry does not recruit beige adipocytes (**Supplementary Fig. 5f**).

Supplementary Fig. 5e UCP1 immunostaining of scWAT of *Mypt1^{flox/flox}* mice (line 2) injected with AAV-mCherry-2A-Cre and acclimated to 12°C for 1 week. The right panel shows high-magnification images. Scale bar, 100 µm. Tissue samples for UCP1 staining were the same as in **Fig. 5d**.

Supplementary Fig. 5f mCherry immunostaining of scWAT of *Mypt1^{flox/flox}* mice (line 2) injected with AAV-mCherry acclimated to 12°C for 1 week. Scale bar, 100 µm.

For original **Fig. 5d** showing *Mypt1^{+flox}::Pdgfra-Cre* mice *in vivo* experiments, we further examined (i) UCP1 immunoblotting (after chronic cold exposure), (ii) histology of scWAT (at room temperature [RT]), and (iii) gene expression profiles (at RT).

(i) Immunoblotting showed that UCP1 protein levels in scWAT were significantly higher than those in control mice when *Mypt1^{+flox}::Pdgfra-Cre* mice were housed at 12 °C for one week (**Fig. 5g**).

Fig. 5g Schematics of the chronic cold exposure to *Mypt1*^{+flox}::*Pdgfra*-Cre mice (upper left). *Mypt1*^{+flox} mice were used as controls. Immunoblotting of mitochondrial fraction of scWATs of *Mypt1*^{+flox}::*Pdgfra*-Cre mice (line 2) acclimated to 12°C for 1 week (lower left). Densitometric quantification of UCP1 immunoblot (right). TOM20 was used for mitochondria loading control. n = 5 per group.

(ii) Histological analysis showed that *Mypt1*^{+flox}::*Pdgfra*-Cre mice had more UCP1-positive multilocular beige adipocytes in scWAT than control mice at RT (**Fig. 5i**).

Fig. 5i Immunohistochemical analysis of scWAT in *Mypt1*^{+flox}::*Pdgfra*-Cre mice (line 2) housed under RT. scWAT were stained with anti-UCP1 antibody.

(iii) qPCR analysis demonstrated that the expression of thermogenic gene mRNA was increased in the scWAT of *Mypt1*^{+flox}::*Pdgfra*-Cre mice even at RT (**Fig. 5h**).

Fig. 5h Expression of thermogenic gene mRNA in scWAT of *Mypt1^{+flox}::Pdgfra-Cre* mice (line 2) housed under RT. Data are mean \pm SEM (*Mypt1^{+flox}* n = 7; *Mypt1^{+flox}::Pdgfra-Cre* n = 6).

With regards to revised **Supplementary Fig. 5i**, results of an *in vivo* experiment using *Mypt1^{+flox}::Adipoq-Cre* mice, gene expression profiling, and histological studies were not performed because no thermogenic phenotypes were obtained, such as increased *Ucp1* expression in scWATs or increased cold tolerance by acute cold exposure (4 °C). These data suggest that JMJD1A does not contribute to the trans-differentiation of mature white adipocytes into beige adipocytes in scWAT.

Reply to major comment 1-2:

We appreciate your valuable comment. Therefore, we immunostained serial sections of scWAT from *Mypt1^{flox/flox}* mice injected with AAV-mCherry-2A-Cre with either UCP1 or mCherry antibody, separately, and examined whether mCherry-2A-Cre positive cells (i.e., MYPT1-depleted cells) overlapped with UCP1 positive cells. The results showed that most of the multilocular adipocytes colocalized with UCP1 and mCherry, whereas mCherry-negative, UCP1-positive beige adipocytes were hardly detected. This indicated that MYPT1 depletion promoted beige adipocyte formation in a cell-autonomous manner (**Fig. 5d**).

Fig. 5d Immunohistochemical analysis of scWAT in *Mypt1^{flox/flox}* mice (line 2) injected with AAV-mCherry-2A-Cre. *Mypt1^{flox/flox}* mice (line 2) were injected with AAV-mCherry-2A-Cre into the inguinal white fat pad after acclimation at 12 °C for 1 week. The scWAT was stained with UCP1 or mCherry antibody. Black arrowheads indicate mCherry-positive and UCP1-negative, i.e., AAV-infected white adipocytes; white arrow heads indicate mCherry- negative and UCP1-negative, i.e., AAV-uninfected white adipocytes; and yellow arrowheads indicate mCherry-positive and UCP-positive, i.e., AAV-infected beige adipocytes. Scale bar, 100 μ m.

The following sentences were added.

“Histological analysis showed that, in the scWAT of *Mypt1^{flox/flox}* mice injected with AAV-mCherry-2A-Cre, expression of UCP1 was induced in multilocular lipid droplets, and many of the UCP1-positive cells co-expressed mCherry-2A-Cre, suggesting that MYPT1 deficiency induced beige adipocytes in mice (**Fig. 5d** and **Supplementary Fig. 5e**). Injection of AAV-mCherry induces mCherry expression in white adipocytes, but this by itself does not recruit beige adipocytes (**Supplementary Fig. 5f**). Reduced MYPT1 protein by AAV-mCherry-2A-Cre injection slightly decreased scWAT weight without affecting body weight (**Fig. 5e** and **5f**).” (Page 15 to page 16)

Major comment 2: Although the authors showed that *Ucp1* expression was regulated by the MYPT1-JMJD1A axis, the effect of MYPT1 deficiency on H3K4me2 is still unclear. They should investigate the levels of H3K4me2 in the *Ucp1* enhancer upon overexpression of demethylation defective JMJD1A mutant or WT-JMJD1A (related to Figs. 3h and 3i)

(*major comment 2-1*). Moreover, the levels of H3K4me2 in *Mypt1*-deficient mice were not determined. Also, they should examine whether the histone methylation of *Ucp1* enhancer is modulated by *JMJD1A* in these experimental conditions (*major comment 2-2*).

Reply to major comment 2-1:

We assume that the reviewer refers to H3K9me2 levels and not H3K4me2 levels, as the former is the substrate for *JMJD1A*. As shown in **Fig. 1n** in the original manuscript, simultaneous depletion of both *MYPT1* (regulatory subunit) and *PP1 β* (catalytic subunit) by siRNAs resulted in higher H3K9me2 levels in the thermogenic gene (i.e., *Ucp1*) in differentiated im-scWAT. However, it is unclear whether this is dependent on the H3K9me2 demethylation activity of *JMJD1A*.

Therefore, to determine whether this change in H3K9me2 is mediated by *JMJD1A*, we examined H3K9me2 levels on the *Ucp1* gene enhancer by overexpression of a demethylation-defective *JMJD1A* mutant (H1120Y) or WT-*JMJD1A*. In im-scWAT overexpressing WT-human *JMJD1A*, H3K9me2 levels in the *Ucp1* enhancer were reduced by *Mypt1* knockdown (**Fig. 3j**). In contrast, in H1120Y-human *JMJD1A*-overexpressing im-scWAT, H3K9me2 levels on the *Ucp1* enhancer were high, and *Mypt1* knockdown hardly reduced H3K9me2 levels. These results indicate that *Ucp1* expression is regulated by the *MYPT1*-*JMJD1A* axis and that the decrease in H3K9me2 levels upon *MYPT1* depletion is dependent on *JMJD1A* demethylation activity.

Fig. 1n H3K9me2 evaluated by ChIP-qPCR in adipocytes differentiated (day 8) from im-scWAT transfected with both si-*Mypt1* #2 and si-*Ppp1cb*#1.

Fig. 3j WT- or H1120Y- hJMJD1A-overexpressing im-scWAT pre-adipocytes were transfected with siRNAs for either control (si-Ctrl) or *Mypt1* (si-*Mypt1*#2). On day 8 of differentiation, cells were harvested and subjected to ChIP-qPCR using an anti-H3K9me2 antibody.

Reply to major comment 2-2:

We agree that the present study did not show the levels of H3K9me2 in scWATs of AAV-Cre-injected *Mypt1*^{fllox/fllox} mice or conditional *Mypt1*-deficient mice. However, it is technically difficult to see a decrease in H3K9me2 at 12 °C for one week of cold stress in *Mypt1* partially deficient mice (**Fig. 1** only for reviewers). This is because, as shown by UCP1 immunohistochemistry (**Fig. 2** only for reviewers), the “beige-ing” of scWAT is much milder at 12 °C than at 8 °C.

Fig. 1 only for reviewers

H3K9me2 ChIP-qPCR in scWATs of *Mypt1*^{fllox/fllox} mice injected with control AAV-mCherry or AAV-mCherry-2A-Cre (4/group). Data are presented as mean ± SEM.

Fig. 2 only for reviewers

Immunohistochemical analysis of scWAT from mice acclimated at 8 °C for two weeks (left) or 12 °C for one week (right). The scWAT was stained with UCP1 antibody.

In addition, scWAT is composed of various cell types, including white adipocytes, beige adipocytes, pre-adipocytes, immune cells, blood cells, and endothelial cells. A recent study reported that adipocytes account for only 50% of all cell types in scWAT (*Cell Reports* Roh et al. 2017). Therefore, it is not easy to see a specific decrease in H3K9me2 on thermogenic genes (e.g., *Ucp1*) by beige-ing because the affected cells comprise only a small fraction of the total cells present in adipose tissue.

We previously reported a slight but significant decrease in H3K9me2 on thermogenic genes in scWAT beige-ing under severe cold stress condition at 4 °C (Abe Y. et al., *Nat Commun* 2018; **Fig. 1f**, see below figure), but in this temperature, many more beige adipocytes were observed. The slight decrease in H3K9me2 in the tissue is due to heterogeneity of the scWAT population, with adipocytes (white and beige) accounting for only 50% of all scWAT cell types, as described above. In addition, beige-ing decreases H3K9me2 in thermogenic genes (e.g., *Ucp1*) but not in other cell types, where H3K9me2 levels are likely high because thermogenic genes are not turned on.

ChIP-qPCR in scWAT from mice placed at 4°C for 1 week (n = 4). The data from Abe Y. et al., 2018 *Nat Commun.* H3K9me2, **Fig. 1f**

Therefore, to evaluate a more homogeneous cell population, we first showed that *Mypt1* depletion reduces H3K9me2 on *Ucp1* gene enhancers and that the MYPT1-JMJD1A axis regulates H3K9me2 on the *Ucp1* enhancers using SVF-derived cultured pre-adipocytes of scWATs. Next, we generated conditional knock-out mice to validate this concept. The knock-out phenotype was consistent with cultured scWAT showing increased thermogenic capacity due to MYPT1 depletion.

Major comment 3: In Fig. 5, what are the levels of JMJD1A and RLC phosphorylation in *Mypt1^{+flox}::Pdgfra-Cre* mice? They should verify whether MYPT1-PP1 β dephosphorylates JMJD1A and RLC *in vivo* in the context of beige adipogenesis during cold exposure.

Reply to major comment 3: We examined JMJD1A phosphorylation in the scWAT of *Mypt1^{+flox}::Pdgfra-Cre* mice acclimated at 12 °C for 1 week. Phosphorylation of JMJD1A was higher in the scWAT of *Mypt1^{+flox}::Pdgfra-Cre* mice than that in control mice (**Fig. 5g**, inset), indicating that MYPT1-PP1 β dephosphorylates JMJD1A *in vivo* in the context of cold-induced beige adipogenesis.

Fig. 5g Immunoblot analysis of P-JMJD1A in tissue homogenates of scWAT of *Mypt1^{+flox}::Pdgfra-Cre* mice after 1 week acclimation to 12°C (inset).

Unfortunately, we were unable to demonstrate changes in RLC phosphorylation *in vivo* under these conditions. RLC phosphorylation levels at 12°C may not be high enough to be detected with antibodies.

Major comment 4: In this study, the authors assumed that *Ucp1* mRNA expression (Figs.

2e, 2h, 2i, 3d, 3h, 3l, 4h) would be the marker of beige adipogenesis. Not only *Ucp1* but also other thermogenic genes and beige marker genes should be investigated.

Reply to major comment 4: We appreciate your valuable comments. We have revised the manuscript to show the expression of not only *Ucp1*, but also other thermogenic genes, in the corresponding figures. These results strengthen our major conclusion that the MYPT1-JMJD1A axis regulates beige adipogenesis.

Supplementary Fig. 2d Thermogenic gene expression in day 8 im-scWAT transfected with si-Ctrl, si-Ctrl+si-Mypt1 #2, or si-Mypt1 #2+si-Mylk.

Supplementary Fig. 2f Changes in the expression of thermogenic genes in im-scWAT adipocytes at day 8 of differentiation overexpressing WT or T18A/S19A human RLC by depletion of *Mypt1* (si-Mypt1 #2).

Supplementary Fig. 2g Effects of thermogenic gene expression by *Mypt1* depletion and blebbistatin treatment during beige adipogenesis. Cultured im-scWATs were transfected with control siRNA (si-Ctrl) or si-*Mypt1* #2, treated with 10 μ M blebbistatin, and induced for differentiation of beige adipocytes. On day 8, cells were harvested for qPCR.

Supplementary Fig. 3e Thermogenic gene expression of im-scWAT transfected with si-Ctrl, si-*Mypt1* #2+si-Ctrl, or si-*Mypt1* #2+si-*Taz* #2 differentiated for beige adipocytes.

Supplementary Fig. 3h Effects of MYPT1-depletion induced thermogenic gene expression on WT and demethylation defective JMJD1A expressing cultured adipocytes. WT- or H1120Y-human JMJD1A-overexpressing im-scWATs were transfected with si-Ctrl or si-Mypt1 #2, differentiated into beige adipocytes, and harvested on day 8 for qPCR.

Supplementary Fig. 3i Restoration of thermogenic gene induction by Mypt1 depletion in catalytically inactive H1120Y-JMJD1A-expressing adipocytes through overexpression of WT-human-JMJD1A. im-scWAT pre-adipocytes expressing JMJD1A-H1120Y that were additionally transduced with either empty

or WT-human JMJD1A lentiviral vector were transfected with si-Ctrl or si-*Mypt1* #2 and induced for beige adipogenesis.

Fig. 4h Effect of Thr696Ala substitution in MYPT1 on thermogenic gene expression during beige adipogenesis. im-scWAT pre-adipocytes transduced with WT- or T696A-human-MYPT1 were transfected with siRNA targeting *Mypt1* (si-*Mypt1* #3) and induced to differentiate into beige adipocytes. On day 8, the cells were harvested, and mRNAs was quantified by qPCR. Data are mean \pm SEM of four biological replicates. *P*-values by Student's t-test.

Major comment 5: *The authors claimed that PKA inhibited MYPT1/PP1beta and, in turn, activated JMJD1A, eventually promoting beige adipogenesis upon cold exposure. Recent studies have shown that turn-off of PKA signaling is involved in beige-to-white transition upon re-warming stimuli (PMID: 29657031, 29692364, 27568548). In this regard, the importance of this study would be further strengthened when they examine whether MYPT1 is involved in beige-to-white transition.*

Reply to major comment 5: We thank the reviewer for valuable comments. Yes, PKA inhibits MYPT1/PP1 β phosphatase activity via T694 phosphorylation of MYPT1. It is possible that MYPT1/PP1 β turns on/off PKA signaling in the reverse process, beige-to-white transition. The original focus of this study was the role of MYPT1 in cold-induced beiging of scWAT and we have already included a very large set of data to investigate this process. Thus, we hope the reviewer would agree that this interesting additional possibility is a separate question that we think would be best examined in a future study.

Major comment 6: *The authors suggested that MYPT1 would play suppressive roles in preadipocytes, not in mature beige adipocytes, which was supported by Adipoq-Cre*

Mypt1 KO mice model (Supplementary Figs. 5f-i). However, given that PKA signaling, JMJD1A, and MYPT1/PPT1beta are also present in mature adipocytes, appropriate explanations of the different roles of MYPT1/PPT1beta in preadipocytes and adipocytes should be provided.

Reply to major comment 6: We thank the reviewer for their valuable comments. As per your comments, MYPT1 exerts its main action on JMJD1A during beige adipogenesis. During this period, JMJD1A erases the repressive H3K9me2 from thermogenic genes, converting silenced chromatin to open chromatin, and thus activating transcription. Phosphorylation of JMJD1A forms a -specific complex with transcription factors and nuclear proteins, such as PPAR, PGC1 α , and PRDM16, to determine the specificity of the target genes (Abe Y et al, *Nat Commun* 2018). Thus, phosphatase MYPT1/PP1 β , which removes this phosphorylation, acts as a repressor during the demethylation period. However, once H3K9me2 is eliminated and beige adipogenesis is completed, JMJD1A is no longer needed and MYPT1/PP1 β no longer exhibits a repressive function.

With regards to *Adipoq-Cre::Mypt1^{lox/lox}* mice, they do not exhibit thermogenic phenotype, suggesting that phospho-JMJD1A does not promote trans-differentiation from white mature adipocytes to beige adipocytes.

In addition, as a preliminary experiment, primary cultured SVFs were prepared from scWATs of *Mypt1^{lox/lox}* mice and infected with adenovirus carrying Cre recombinase (Adeno-Cre) on the day before (Day 1; pre-adipocyte stage) or seven days after (Day 7; mature adipocyte stage) differentiation to eliminate MYPT1. After differentiation, adipocytes were harvested on day 8 and thermogenic gene expression was examined. The results (**Fig. 3 only for reviewers**) showed that the expression of thermogenic genes was higher in cells infected on day 1 than in those infected on day 7, suggesting that *Mypt1* depletion does not affect the transcription of thermogenic genes in mature beige adipocytes, but only early in differentiation. We acknowledge that this is not a perfect experiment, and there are limitations in the interpretation; however, the results are consistent with the model and it will require significant future studies to clarify these points.

Fig. 3 only for reviewers

Primary preadipocytes isolated from SVF of scWAT of *Mypt1^{lox/lox}* mice were infected with either AxCAN-LacZ (adeno-LacZ) or AxCAN-Cre (Adeno-Cre) that expresses the LacZ and Cre recombinase gene, respectively, under the control of the CAG promoter for 24 h on one day before differentiation (Day -1 infection) or on day 7 (Day 7 infection) during beige adipogenesis. mRNA levels of thermogenic genes, general adipogenic genes, and *Mypt1* on day 8 of differentiation were quantified by RT-qPCR. Data are mean \pm SEM of three technical replicates.

Minor comments

Minor comment 1: Many graphs do not display *n* numbers and statistics.

Reply to comment 1: We apologize that some of the graphs do not display sample numbers and statistics in the original manuscript. We have added sample numbers and statistics to all figures (Figs. 1i, 1j, 1k, 1n, 2e, 2h, 2i, 3d, 3h, 3i, and **Supplementary Figs. 1b, 1c, 1d, 1e, 1f, 2c, 2e, 3b, 3d, 5g, and 5m** in the revised manuscript) and their legends.

The following sentences are added to each figure legend.

“**d, e, f, i, j, k, l, n** Representative of three (**e, f, i, j, k, l**) or two (**d, n**) independent experiments. Data are presented as the mean \pm SEM of three technical replicates in **i, j, k, and n**. Representative data are presented in **Supplementary Table 1. n** Unpaired two-tailed Student’s t-test. **i, j, and k** One-way ANOVA with Tukey’s multiple comparisons test.” (Page 57 to 58, **Fig. 1** legend of revised manuscript).

“**e, f, g, i** Representative of three (**e, i**) or two (**f, g**) independent experiments. Data are mean \pm SEM of three technical replicates in **e** and **i**. **e, h, i** One-way ANOVA with Tukey’s multiple comparisons test.” (Page 59, **Fig. 2** legend of revised manuscript).

“**d, h, i, j** Representative of three (**h**) or two (**d, i, j**) independent experiments. Data are mean \pm s.e.m. of three technical replicates in **d, h, i, and j**. **d, h, i, j** One-way ANOVA with Tukey’s multiple comparisons test.” (Page 61, **Fig. 3** legend of revised manuscript).

“**a, b, c, d, e, f** Representative of three (**b, e**) or two (**a, c, d, f**) independent experiments. Data are mean \pm SEM of three technical replicates in **b, c, d, e, and f**. **c** and **f** Unpaired two-tailed Student’s t-test. **b, d, and e** One-way ANOVA with Tukey’s multiple comparisons test.” (**Supplementary Fig. 1** legend of revised manuscript).

“**c, d, e, g** Representative of two independent experiments. Data are mean \pm SEM of three technical replicates in **c, d, e, and g**. **c, d, e, f, and g** One-way ANOVA with Tukey’s multiple comparisons test.” (**Supplementary Fig. 2** legend of revised manuscript).

“**b, d, e, h, i** Representative of three (**b**) or two (**d, e, h, i**) independent experiments. Data are mean \pm SEM of three technical replicates in **b, d, e, h, and i**. **b** Unpaired two-tailed Student’s t-test. **d, e, h, and i** One-way ANOVA with Tukey’s multiple comparisons test.” (**Supplementary Fig. 3** legend of revised manuscript).

“**g** and **m** representative of two independent experiments. Data are mean \pm SEM of three technical replicates in **g** and **m**. **g, k, l, m, n, o, p, and q** Unpaired two-tailed Student’s t-test. Source data are provided as a Source data file.” (**Supplementary Fig. 5** legend of revised manuscript).

***Minor comment 2:** In Figs. 1m and 3e, how the relationship between the binding site of JMJD1A and gene expression was investigated? What is the rationale for setting the standard to 50 kb?*

Reply to comment 2: To examine the relationship between the binding sites of JMJD1A and gene expression, we defined JMJD1A-regulated genes as those that met the following criteria: expression was induced more than 1.5-fold or less than 1/1.5-fold and localized within 50 kb from JMJD1A binding sites, as described under “Methods”. The score for each gene (JMJD1A binding) was defined as the sum of the

scores of the peaks located in the ± 50 kb region around its TSS. A higher score indicated stronger binding of JMJD1A to the promoter or enhancer region of the gene of interest.

We used a setting of 50 kb according to the following published papers: (1) *Nat Cell Biol* 2017 [10.1038/ncb3590]; (2) *Genome Research* 2015 [10.1101/gr.188300.114]; (3) *Epigenetics & Chromatin*, 2020 [10.1186/s13072-020-0327-0].

Minor comment 3: For the MS/MS data reports (Figs. 1b, 1c), more details such as peptide numbers and protein match percentage should be included

Reply to minor comment 3: According to the comment, a table showing peptide numbers and protein match rates of the MS/MS data has been included as “Supplemental Data 1” in the revised manuscript.

Identified protein	Molecular Weight (kDa)	Peptide numbers				Protein match percentage (%)			
		Experiment 1		Experiment 2		Experiment 1		Experiment 2	
		Empty	FLAG-JMJD1A	Empty	FLAG-JMJD1A	Empty	FLAG-JMJD1A	Empty	FLAG-JMJD1A
KDM3A_MOUSE	148 kDa	0	35	0	74	0	18.8	0	28.3
ST5_MOUSE	127 kDa	0	6	0	0	0	7.41	0	0
CO1A1_MOUSE	138 kDa	0	3	0	0	0	3.79	0	0
KRT82_MOUSE	57 kDa	0	3	9	0	0	4.07	6.4	0
LRRF1_MOUSE	79 kDa	0	3	0	0	0	6.72	0	0
RS13_MOUSE	17 kDa	0	3	0	0	0	26.5	0	0
CYTSA_MOUSE	124 kDa	3	14	0	2	2.86	16	0	2.68
TPM1_MOUSE	33 kDa	11	41	0	15	29.2	53.5	0	40.8
C1TM_MOUSE	106 kDa	3	10	0	1	3.58	9.21	0	1.23
MYPT1_MOUSE	115 kDa	5	16	0	6	3.69	17.8	0	7.39
FBLN2_MOUSE	132 kDa	1	3	0	0	1.06	3.77	0	0
PP1B_MOUSE	37 kDa	1	3	0	1	3.98	12.2	0	3.98

Supplementary Data 1 (A part is shown here)

Minor comment 4: The interpretation of the oil-red O experiment is puzzling. It was not fully explained in the text.

Reply to minor comment 4: We thank the reviewer for this important comment. This is related to the following comment by reviewer #2: “This separates current study from many other reports that claim protein X or Y are thermogenic fat regulators but most likely they just affect fat function in general.”

To show that *Mypt1* knockdown (or other settings) did not affect lipid accumulation (i.e., general adipogenesis), but did affect beige adipogenesis (thermogenic gene expression), we presented the results of Oil Red O staining, but this point was not clearly explained in the original paper. Therefore, we have added the following sentence to the revised manuscript:

"Oil red O staining showed that lipid accumulation (i.e., general adipogenesis) is not affected by MYPT1 depletion (**Fig. 1i**, inset), indicating MYPT1 regulates specific thermogenic genes during adipogenesis." (Page 8, lines 9–11)

Minor comment 5: In Figs.5e, 5f, 5g, the number of biological replicates appears to be small.

Reply to minor comment 5: We agree with the reviewer's concern. Shown in **Fig. 5f** and 5g in the original manuscript (acute cold exposure), there were three biological replicates; therefore, we omitted these experiments. Alternatively, we analyzed *Mypt1^{+floxed}::Pdgfra*-Cre mice after chronic (1 week) cold exposure at 12 °C or RT. After cold exposure, UCP1 protein levels in scWAT were significantly higher than those in control mice (**Fig. 5g** in the revised manuscript). In addition, *Mypt1^{+floxed}::Pdgfra*-Cre mice showed higher expression of thermogenic genes and UCP1-positive multilocular beige adipocytes than control mice at RT (**Fig 5h, 5i** in the revised manuscript). These results indicate that MYPT1 depletion in pre-adipocytes promotes beige adipogenesis.

The acute cold tolerance test in the original **Fig. 5e** was performed under RT, where beige adipogenesis was significantly promoted in *Mypt1^{+floxed}::Pdgfra*-Cre mice compared to that in control mice. Body temperature change in control mice, although the number of biological replicates was small, was comparable to that in control mice in **Supplementary Fig. 5j**. In contrast, knockout mice (n = 6) were more resistant to cold stress compared to control mice, shown in **Fig. 5i**. Thus, we considered that the data are promising and have kept this data in **Supplementary Fig. 5q** in the revised manuscript.

In addition, we included metabolic data from mice of this genotype fed a high-fat diet (HFD) to confirm the higher energy expenditure phenotype. HFD feeding resulted in higher body weight gain in the control (*Mypt1^{+floxed}*) group, while the *Mypt1^{+floxed}::Pdgfra*-Cre mice (male, n = 20–22), which exhibited less body weight gain in comparison(**Fig. 5j**). These mice had improved glucose tolerance and reduced plasma insulin levels, showing an improved metabolic phenotype when challenged with HFD (**Fig. 5k, 5i**, and **Supplementary Fig. 5r**). These data alternatively support a higher thermogenic/energy consumption phenotype in *Mypt1^{+floxed}::Pdgfra*-Cre mice. We have removed the original **Fig. 5f** and **5g** from the revised manuscript. We hope that the reviewer agrees that these additions and removals increased the overall quality of the study presented here.

The following sentences were added.

“When *Mypt1*^{+/*fl*ox}::*Pdgfra*-Cre mice were housed at 12 °C for 1 week, UCP1 protein levels in the scWAT were significantly higher than those in control mice (**Fig. 5g**). Phosphorylation of JMJD1A was higher in the scWAT of *Mypt1*^{+/*fl*ox}::*Pdgfra*-Cre mice than in that of control mice (**Fig. 5g**, inset). Expression of thermogenic genes was also increased in the scWAT of *Mypt1*^{+/*fl*ox}::*Pdgfra*-Cre mice, even in RT housing (**Fig. 5h**). Histological analysis showed that *Mypt1*^{+/*fl*ox}::*Pdgfra*-Cre mice had more UCP1-positive multilocular beige adipocytes in scWAT than did control mice acclimated at RT (**Fig. 5i**).” (Page 17, lines 5-11)

“In addition, HFD feeding at RT resulted in higher body weight gain in control mice, while *Mypt1*^{+/*fl*ox}::*Pdgfra*-Cre mice exhibited reduced body weight gain in comparison (**Fig. 5j**). Furthermore, *Mypt1*^{+/*fl*ox}::*Pdgfra*-Cre mice showed improved glucose tolerance and lower serum insulin levels under fasting conditions or after glucose injection compared with control mice (**Fig. 5k, 5l, Supplementary Fig. 5r**), indicating that MYPT1 depletion is associated with a higher energy consumption phenotype and improved glucose metabolism. These results indicate that MYPT1 is a crucial regulator of thermogenic capacity in scWAT and energy metabolism *in vivo*.” (Page 17, line 14 to Page 18, line 2)

Fig. 5g Schematics of the chronic cold exposure to *Mypt1*^{+/*fl*ox}::*Pdgfra*-Cre mice (upper left). *Mypt1*^{+/*fl*ox} mice were used as controls. Immunoblotting of mitochondrial fraction of scWATs of *Mypt1*^{+/*fl*ox}::*Pdgfra*-Cre mice (line 2) acclimated to 12°C for 1 week (lower left). Densitometric quantification of UCP1 immunoblot (right). TOM20 was used for mitochondria loading control. n = 5 per group.

Fig. 5h Expression of thermogenic gene mRNA in scWAT of *Myt1^{+flox}::Pdgfra-Cre* mice (line 2) housed under RT. Data are mean ± SEM (*Myt1^{+flox}* n = 7; *Myt1^{+flox}::Pdgfra-Cre* n = 6).

Fig. 5i Immunohistochemical analysis of scWAT in *Myt1^{+flox}::Pdgfra-Cre* mice (line 2) housed under RT. scWAT were stained with anti-UCP1 antibody.

Fig. 5j Changes of body weight (left) and body weight gain (right). *Myt1^{+flox}* (n = 16) and *Myt1^{+flox}::Pdgfra-Cre* (n = 18) mice (line 2) were fed on HFD for 15

weeks under RT.

Fig. 5k Glucose tolerance test (GTT) in each genotype group mice fed on HFD in **j** (*Mypt1*^{+/*flox*}: n = 16; *Mypt1*^{+/*flox*}::*Pdgfra*-Cre: n = 17) (line 2).

Fig 5l Fasting insulin levels in each genotype group mice fed on HFD in **j** (*Mypt1*^{+/*flox*}: n = 14; *Mypt1*^{+/*flox*}::*Pdgfra*-Cre: n = 14) (line 2).

Supplementary Fig. 5r Glucose stimulated insulin secretion (GSIS) in each genotype group mice fed on HFD in **Fig. 5j** (*Mypt1*^{+/*flox*}: n = 14; *Mypt1*^{+/*flox*}::*Pdgfra*-Cre: n = 14) (line 2). The data at 0 min were the same as those

in **Fig. 5l**. Paired t-test for comparison of glucose effect. Unpaired Student's t-test for comparison of groups.

Reply to minor comment 6: As in our response to minor comment 1, we have provided the number of biological replicates for all experiments in the figure legends in the revised manuscript. In addition, we provide a table showing detailed information for all mouse experiments (number, sex, and age) in **Supplemental Table 5** in the revised manuscript.

Minor comment 7. The western blot in Figs. 1d and 4g were performed individually, suggesting either inconsistent or incomparable. These blots should be reperformed in the same blot.

Reply to minor comment 7:

We repeated the co-immunoprecipitation analysis of **Fig. 1d** and confirmed an interaction between MYPT1 and JMJD1A. We also repeated the P-JMJD1A immunoblot analysis of **Fig. 4g** and confirmed that ISO-induced JMJD1A phosphorylation was markedly decreased in T696A-MYPT1 cells. The new data are shown in **Fig. 1d** and **4g** in the revised manuscript.

Fig. 1d Co-immunoprecipitation of MYPT1 and P-JMJD1A in NIH-3T3 cells. Cell lysates from NIH-3T3 cells at the indicated time points after an IBMX-containing differentiation cocktail were immunoprecipitated with anti-JMJD1A antibody and subjected to SDS-PAGE. Immunoblotting was performed using anti-P-JMJD1A antibody. Subsequently, MYPT1 immunoblotting was performed without antibody stripping, resulting in the detection of both MYPT1 and P-JMJD1A (arrowhead).

Fig. 4g Immunoblotting with anti-P-JMJD1A or anti-mouse JMJD1A following immunoprecipitation with anti-mouse JMJD1A from *WT*- or T696A-human MYPT1-transduced im-scWAT treated with ISO (10 μ M) for 0, 20, and 40 min on day 0 transfected with si-*Mypt1* #3.

In addition, we want to emphasize that the original data shown in **Fig. 1d** and **4g** were not performed individually, but on the same blots (**Fig. 4** and **Fig. 5** only for reviewers). Irrelevant samples on the gels were eliminated. We apologize for the lack of uncropped images, causing ambiguity, in the initial submission.

Original Fig. 1d

Fig. 4 only for reviewers Uncropped images for original **Fig. 1d**.

Original Fig. 4g

Fig. 5 only for reviewers Uncropped images for original **Fig. 4g**.

Minor comment 8. In place of bar graphs, small datasets show the full data as univariate scatter plots that also show mean and standard deviation (Fig. 1i, 1j, 1k, 2e, 2i, 3d, 3h, 3i).

Reply to minor comment 8: All bar graphs have been replaced with scatter plots with means and standard errors (**Fig. 1i, 1j, 1k, 2e, 2i, 3d, 3h, 3i, Supplementary Figs. 1b, 1d, 1e, 1f, 2c, 2e, 3d, 5g, and 5m** in the revised manuscript).

Reviewer #2: *The manuscript titled “MYPT1-PP1B phosphatase negatively regulates both chromatin landscape and co-activator recruitment to influence gene expression during beige adipogenesis” by Takahashi et al reported a previously unknown regulator MYPT1-PP1B in beige fat biology. The group has previously reported that a histone H3 lysine 9 (H3K9) demethylase JMJD1A plays an important role regulating beige fat function, and is activated through phosphorylation at serine 265 by protein kinase A (Nature Communications 2015, Abe et al. and Nature Communications 2018 Abe et al). Current study is a nice follow-up to reveal the identity of the phosphatase that is responsible for the dephosphorylation of s265 in JMJD1A in beige fat. The unbiased experimental approach was well controlled and presented data were convincing. In addition, the author further uncovered an interesting connection between MYPT1-PP1B to myosin regulatory light chain (RLC) regulated pathway, which mediates beige fat function through TAZ/YAP. Even though this is continuation of previous study, the novelty and significance of the present study is evident.*

The manuscript is well written and most of the results are of high quality, strongly supporting proposed hypothesis. In particular, the authors demonstrated that general adipogenesis is not affected by MYPT1-PP1B, rather its effects are specifically affecting thermogenic function. This separates current study from many other reports that claim protein X or Y are thermogenic fat regulators but most likely they just affect fat function in general.

Major comments

*This reviewer does have some comments/suggestions for data shown in figure 5, in particular the data using *Mypt1^{+/-flox};pdgfra-cre* mice.*

*As of right now, there is no perfect Cre line for general preadipocytes, nor for beige fat precursor. PDGFRA-CRE has been used as preadipocyte cre line in many studies, but its caveats were well documented. Indeed, the fact that *Mypt1^{flox/flox};pdgfra-cre* mice die embryonically suggests that MYPT1 in other cell types in addition to preadipocyte may play an essential role in development. The drastic phenotype shown in fig5E and fig5g from heterogeneous *Mypt1* in PDGFRA-lineage is a bit complicated to interpret. The most responsive organs upon acute cold exposure are skeletal muscle which regulates shivering and classical brown fat which will be immediately activated upon cold. In comparison, beige adipogenesis within subcutaneous fat upon cold exposure, the full extent of activation may take a long time and involve tissue remodeling, including new precursor for beige adipocytes to differentiate. The fact that after only 8 hrs, there is body weight change and subcutaneous adipose tissue mass change and body temperature change, it*

is unlikely caused by half dose of mypt1 in just beige precursors. Further exploration should be carried out. Alternatively, the authors may consider to just remove this model from the current study, since there are several other in vivo models, which are better controlled with fewer caveats and also support the hypothesis.

Reply to major comment

We appreciate reviewer's thorough understanding of our manuscript.

Firstly, we will address the concern “*In comparison, beige adipogenesis within subcutaneous fat upon cold exposure, the full extent of activation may take a long time and involve tissue remodeling, including new precursor for beige adipocytes to differentiate. The fact that after only 8 hrs, there is body weight change and subcutaneous adipose tissue mass change and body temperature change, it is unlikely caused by a half dose of mypt1 in just beige precursors*”

We would like to emphasize that we do not think that beige-ing of *Mypt1*^{fllox/+} mice occurred within 8 h of acute cold exposure, but through rearing at RT. We also need to emphasize that the data in the original **Fig. 5g** show no significant change in body weight or subcutaneous adipose tissue mass after 8 h of cold exposure between the two groups ; therefore, beige-ing did not occur within this short time period.

As the reviewer may know, RT is lower than the thermoneutral temperature, 30 °C, so an intermediate cold stimulus is present. Beige-ing occurs via sympathetic nerve activity. Beige-ing was promoted in scWATs of mice reared at RT; *Pdgfr-Cre::Mypt1*^{fllox/+} mice showed a higher expression of thermogenic genes associated with beige-ing than the control group at RT, as represented by *Ucp1*, *Cpt1b*, and *Ppara* ($P < 0.05$) (**Fig. 5h**). Furthermore, *Mypt1*^{+fllox::Pdgfra-Cre} mice had more UCP1-positive multilocular beige adipocytes in scWAT than control mice at RT (**Fig. 5i**).

It has been reported that mice with advanced beige-ing become resistant to acute cold (Ikeda, K. et al. *Nat Med* 23, 1454-1465, 2017). Regarding *Pdgfra-Cre*, recent studies have shown that *Pdgfra* is expressed in progenitor adipocytes, but not in skeletal muscle or liver (Saavedra-Peña, RDM. et al. *J Mol Endocrinol* 68, 179-194, 2022); however, it is expressed in glial cells and gonads (*Journal of Molecular Endocrinology* 68, 179-194, 2022, DOI: <https://doi.org/10.1530/JME-21-0260>).

In addition, as in our response to minor comment 5 from reviewer #1, we would like to provide additional metabolic data regarding *Mypt1*^{fllox/+::Pdgfra-Cre} mice that have reduced body weight gain, better glucose tolerance, and lower plasma insulin upon

increased glucose challenging. These data alternatively support a higher thermogenic/energy consumption phenotype in *Mypt1^{+floX}::Pdgfra*-Cre mice (Figs. 5j, 5k, and 5l). We hope that we could provide a satisfactory response to your constructive criticism.

Fig. 5h Expression of thermogenic gene mRNA in scWAT of *Mypt1^{+floX}::Pdgfra*-Cre mice (line 2) housed under RT. Data are mean \pm SEM (*Mypt1^{+floX}* n = 7; *Mypt1^{+floX}::Pdgfra*-Cre n = 6).

Fig. 5i Immunohistochemical analysis of scWAT in *Mypt1^{+floX}::Pdgfra*-Cre mice (line 2) housed under RT. scWAT were stained with anti-UCP1 antibody.

Fig. 5j Changes of body weight (left) and body weight gain (right). *Mypt1*^{+/*flox*} (n = 16) and *Mypt1*^{+/*flox*}::*Pdgfra-Cre* (n = 18) mice (line 2) were fed on HFD for 15 weeks under RT.

Fig. 5k Glucose tolerance test (GTT) in each genotype group mice fed on HFD in **j** (*Mypt1*^{+/*flox*}: n = 16; *Mypt1*^{+/*flox*}::*Pdgfra-Cre*: n = 17) (line 2).

Fig 5l Fasting insulin levels in each genotype group mice fed on HFD in **j** (*Mypt1*^{+/*flox*}: n = 14; *Mypt1*^{+/*flox*}::*Pdgfra-Cre*: n = 14) (line 2).

Minor point 1: *In data availability section, the authors indicate that the RNA-seq transcriptome data and JMJD1A ChIP-seq data have been deposited at GEO but the accession number was listed as “GSEXXXXX”. Customarily, most authors deposit these data as private data protected with a password, with public release set upon publication date of the manuscript. An actual GSE number should be included, and a password should be made available to reviewers upon request.*

Reply to minor point 1: We apologize that the accession numbers of RNA-seq transcriptome data and JMJD1A ChIP-seq data were not provided in the original manuscript. The GSE number (GSE202506) has been included in the revised manuscript. A password for accessing the data will be provided to reviewers upon request.

The following sentence was added.

“RNA-seq transcriptome data and JMJD1A ChIP-seq data (day 0 and day 4) were deposited in the Gene Expression Omnibus (GEO) database with accession numbers GSE202506.” (Page 48, lines 1-2)

REVIEWERS' COMMENTS

Reviewer #1 (Remarks to the Author):

All the concerns are properly addressed by the authors. This reviewer has no further questions for this revised ms.

Reviewer #2 (Remarks to the Author):

The authors did a thorough revision and the updated manuscript has been further improved.